# An avalanche-and-surge robust ultrawide-bandgap heterojunction for power electronics

Feng Zhou[1,5], Hehe Gong[1,5], Ming Xiao[2,5], Yunwei Ma[2], Zhengpeng Wang[1], Xinxin Yu[1], Li Li[3,4], Lan Fu[4], Hark Hoe Tan[3,4], Yi Yang[1], Fang-Fang Ren[1], Shulin Gu[1], Youdou Zheng[1], Hai Lu[1]✉, Rong Zhang[1]✉, Yuhao Zhang[2]✉ & Jiandong Ye[1]✉

Avalanche and surge robustness involve fundamental carrier dynamics under high electric field and current density. They are also prerequisites of any power device to survive common overvoltage and overcurrent stresses in power electronics applications such as electric vehicles, electricity grids, and renewable energy processing. Despite tremendous efforts to develop the next-generation power devices using emerging ultra-wide bandgap semiconductors, the lack of effective bipolar doping has been a daunting obstacle for achieving the necessary robustness in these devices. Here we report avalanche and surge robustness in a heterojunction formed between the ultra-wide bandgap n-type gallium oxide and the wide-bandgap p-type nickel oxide. Under 1500 V reverse bias, impact ionization initiates in gallium oxide, and the staggered band alignment favors efficient hole removal, enabling a high avalanche current over 50 A. Under forward bias, bipolar conductivity modulation enables the junction to survive over 50 A surge current. Moreover, the asymmetric carrier lifetime makes the high-level carrier injection dominant in nickel oxide, enabling a fast reverse recovery within 15 ns. This heterojunction breaks the fundamental trade-off between robustness and switching speed in conventional homojunctions and removes a key hurdle to advance ultra-wide bandgap semiconductor devices for power industrial applications.

Power devices are essential building blocks for high-efficiency energy conversion in power electronics systems. The market size of power semiconductor devices has reached US$40 billion driven by applications like electric vehicles, data centers, electric grids, and renewable energy processing[1]. Deployment of new semiconductors is a fundamental driving force to advance power electronics. The last decade witnesses the success of wide bandgap (WBG) semiconductors, e.g.,

gallium nitride (GaN) and silicon carbide (SiC)[2–5]. On the horizon, ultrawide bandgap (UWBG) semiconductors hold tremendous promises for the next-generation power electronics[6–9].

Power devices operate as switches between the high blocking voltage and high conduction current. Their robustness against overvoltage and overcurrent stresses is as important as their performance under normal operations. Such robustness is crucial for any power

[1]School of Electronic Science and Engineering, Nanjing University, 210008 Nanjing, China. [2]Center for Power Electronics Systems, Virginia Polytechnic Institute and State University, Blacksburg 24060 VA, USA. [3]Australian National Fabrication Facility ACT Node, The Australian National University, Canberra ACT 2601, Australia. [4]ARC Centre of Excellence for Transformative Meta-Optical Systems, Department of Electronic Materials Engineering, Research School of Physics, The Australian National University, Canberra ACT 2600, Australia. [5]These author contributed equally: Feng Zhou, Hehe Gong, Ming Xiao. ✉e-mail: hailu@nju.edu.cn; rzhang@nju.edu.cn; yhzhang@vt.edu; yejd@nju.edu.cn

device, as they allow devices to temporarily survive the common faults in power systems, e.g., short circuit, excessive load, arc/ground faults, before the protection circuitry intervenes[10]. Avalanche is the desirable mechanism of power devices to withstand overvoltage stresses, as it allows them to accommodate high avalanche current ($I_{AVA}$) at the avalanche breakdown voltage ($BV_{AVA}$) and thus dissipate the excessive energy in circuits[11]. The avalanche and surge current capabilities usually represent the power device robustness against electrical and electrothermal overstress[12].

The native p-n junction is the enabling device structure for avalanche and surge robustness in silicon (Si), silicon carbide (SiC), and gallium nitride (GaN) devices[12,13], the three power semiconductor technologies have reached commercialization. The avalanche hinges on the impact ionization (I. I.) and multiplication occurring at the junction, as well as the efficient removal of the I. I.-produced non-equilibrium carriers. Under forward bias, the capability to withstand high surge current relies on the high-level carrier injection across the p-n junction, which reduces the device resistance as well as suppresses the ramp-up of power loss and junction temperature[12]. In contrast to homojunctions, avalanche or surge current robustness has not been demonstrated in heterojunction-based power devices to date. Only a few avalanche-capable heterojunctions were reported in low-voltage, low-power optoelectronic devices, and the avalanche-like features observed in some unipolar heterostructures are still in debate[14].

Compared to Si and WBG counterparts, UWBG power devices possess superior performance limits[1,8]. However, native p-n homojunction is difficult to form in UWBG materials due to the challenges of achieving efficient bipolar doping[15]. This fundamentally limits the robustness of UWBG power devices, of which Gallium oxide ($Ga_2O_3$) is an example[16–18]. Benefitting from its high critical electric field, controllable n-type doping, and the large-area wafer availability, $Ga_2O_3$ power devices are advancing fast towards applications[19,20], Whereas, due to the flat valence band and strong self-trapping of holes, the reliable p-type doping in $Ga_2O_3$ is very challenging, although p-type $Ga_2O_3$ has been reported by some group[21–23]. As an alternative, heterojunctions between $Ga_2O_3$ and foreign p-type oxide, e.g., nickel oxide (NiO) or copper oxide[24], have recently been deployed in the design of $Ga_2O_3$ bipolar power devices[15]. Despite excellent device performance, the viability of avalanche and surge robustness in such heterojunctions remains a fundamental knowledge gap. Meanwhile, the impact of band discontinuity on carrier transport is largely unexplored under the high electric field (E-field), high current density, and fast switching conditions.

This work fills this gap by demonstrating avalanche and surge robustness in NiO/$Ga_2O_3$ p-n heterojunctions through device innovations and circuit characterizations, whilst relevant carrier dynamics are also revealed through microscopic techniques and physics-based simulations. Large-area NiO/$Ga_2O_3$ p-n heterojunction diodes (HJDs) with advanced edge terminations are designed and fabricated, followed by the avalanche and surge circuit tests complying with industrial standards. Subsequently, the electron beam-induced current (EBIC) characterization and simulations reveal the carrier transport dynamics under critical avalanche and surge conditions. As a key enabler for surge robustness, the bipolar conductivity modulation is found to be dominantly in the NiO with a high hole concentration, while it is usually in the lightly-doped side in conventional homojunctions. This distinction allows NiO/$Ga_2O_3$ heterojunctions to simultaneously achieve a smaller reverse recovery and higher switching speed with robustness comparable or superior to that of conventional homojunction.

## Results

### NiO/$Ga_2O_3$ p-n heterojunction diode

For power devices, edge termination design is critical to control the E-field crowding, avoid premature breakdown, and access the device

$BV_{AVA}$. Here we employ an edge termination that combines small-angle beveled junction termination extension (JTE) and a high-k field plate. Figure 1a presents the three-dimensional schematic diagram of the large-area (3 mm × 3 mm) NiO/$Ga_2O_3$ p-n HJD fabricated on 2-inch free-standing $Ga_2O_3$ wafers. The p-type region consists of lightly- and heavily-doped NiO layers (i.e., p-NiO and $p^+$-NiO). The p-NiO layer can reduce the leakage current and favor the JTE design. At the device edge, the p-NiO extension functions as a JTE, and the small beveled angle allows for a gradual decrease in charge density away from the active region, which continuously reduces the depletion curvature and surface E-field[25]. The high-k field plate conformally covers the NiO JTE and can further passivate the peak E-field.

The HJD fabrication starts with the deposition of NiO films on n-$Ga_2O_3$ drift layer via CMOS-compatible RF magnetron sputtering technique. The hole concentrations in NiO are modulated by tuning the gas flux ratios of Ar/$O_2$ in the sputtering process, resulting in a hole concentration of $5.8 \times 10^{17}$ cm$^{-3}$ and $2.9 \times 10^{19}$ cm$^{-3}$ in p-NiO and $p^+$-NiO layers, respectively. The beveled angle in NiO is implemented by adjusting the gap between the shadow mask and $Ga_2O_3$ wafer as well as the declination angle of the NiO target in the sputtering process, as detailed in Supplementary Section S1. Amorphous barium titanate (BaTiO$_3$), a perovskite oxide with an ultrahigh dielectric constant, is deposited by RF sputtering between the anode metal and NiO to form the field plate. In addition to bare-die devices, some HJDs are sealed in TO-220 packages for circuit tests. More detailed fabrication and packaging process is described in Methods, Supplementary Section S1, and Supplementary Movie 1. Figure 1b shows the cross-sectional scanning transmittance electron microscopy (STEM) image of the edge termination, revealing a bevel angle of 11°. The high-resolution TEM image in Fig. 1c shows an atomically sharp interface of the NiO/$Ga_2O_3$ junction with excellent lattice alignments of the (111)-oriented NiO with (001) $Ga_2O_3$.

TCAD simulations are performed to investigate the heterojunction band structure and the device E-field management. The simulation models are detailed in Supplementary Section S4. As shown in Fig. 1d, at equilibrium, the NiO/$Ga_2O_3$ junction exhibits a type-II (staggered) band alignment with the conduction band and valence band offsets being 2.1 and 3.2 eV, respectively. Figure 1e shows the simulated E-field contours in the HJD with only the NiO JTE and full termination, both at a reverse bias of 1600 V. The high-k field plate shifts the peak E-field away from the junction edge and lowers the peak E-fields in $Ga_2O_3$ and NiO (from 6.61 MV/cm to 4.57 MV/cm and from 7.91 MV/cm to 2.62 MV/cm, respectively). As a result, a nearly uniform E-field is present at the NiO/$Ga_2O_3$ junction, enabling a uniform and robust avalanche.

Capacitance-voltage (C–V) characterizations are performed for the HJD at frequencies ranging from 1 kHz to 1 MHz, showing negligible frequency dispersion at various biases (Fig. 1f). This indicates the presence of minimal interface states at the heterojunction. The net donor concentration in the $Ga_2O_3$ drift layer and the built-in potential of the heterojunction are also extracted from the C–V characteristics to be $1.7 \times 10^{16}$ cm$^{-3}$ and 2.1 V, respectively (see Supplementary Section S2). From the built-in potential and band offsets, the barrier heights for electrons and holes are 4.2 and 5.3 eV, respectively. This suggests the electron injection could be more pronounced than hole injection at large forward biases.

### Avalanche breakdown robustness

Avalanche breakdown is desirable for both power devices and power electronics systems. For devices, it allows for a non-destructive breakdown with a positive temperature coefficient of $BV_{AVA}$[12] and a smaller overvoltage margin as required for a certain voltage rating[26]. For systems, the concurrence of high $I_{AVA}$ and high $BV_{AVA}$ can dissipate the surge energy and prevent it from further circulating in the circuitry[11]. The avalanche characterization should cover all these signatures. Here we employ the quasi-static current-voltage (I–V) characterization to

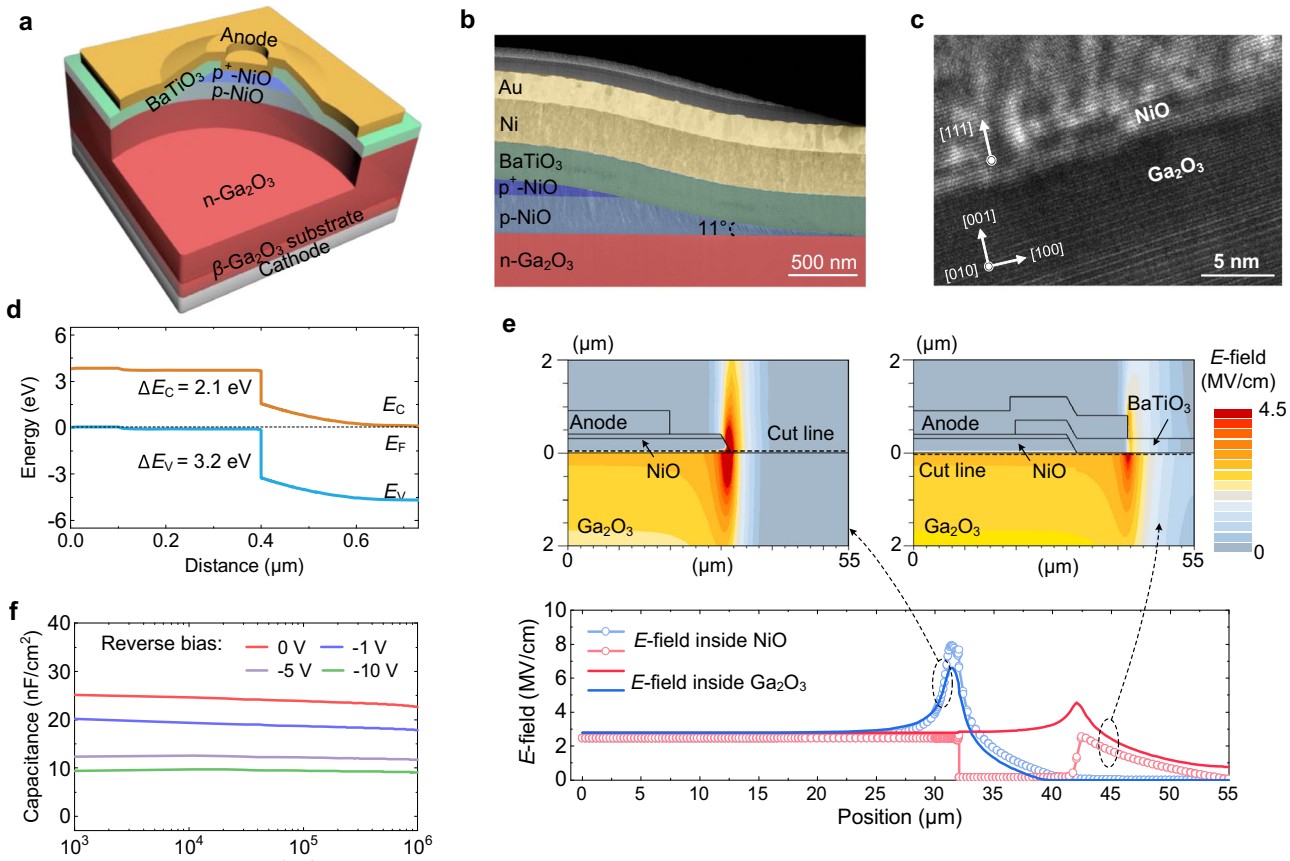

**Fig. 1 | NiO/Ga₂O₃ p-n heterojunction power device. a** Three-dimensional schematic of the NiO/Ga₂O₃ HJD, showing the double-layered NiO (a 300-nm-thick p-NiO and a 100-nm-thick p⁺-NiO) and a high permittivity BaTiO₃ dielectric layer with 11° beveled-mesa termination. **b** STEM image of the termination region of the heterojunction in false color to highlight different layers. **c** High-resolution cross-section TEM image of the heterojunction interface. **d** Schematic energy band diagram of the HJD at zero bias. **e** Simulated in-plane *E*-field contour of devices with and without the BaTiO₃ dielectric layer. **f** Frequency-dependent capacitance characteristics measured at different reverse bias voltages.

probe the $BV_{AVA}$ behaviors and the unclamped inductive switching (UIS) circuit to measure the $I_{AVA}$ and avalanche energy ($E_{AVA}$)[27].

Figure 2a shows the reverse *I–V–T* characteristics of the NiO/Ga₂O₃ HJD at temperatures (*T*) from 25 to 175 °C. The reverse current is low (<0.2 μA) and shows weak bias dependence below 1500 V. Then it rises sharply due to the initiation of I. I. and multiplication. The $BV_{AVA}$ increases from 1545 to 1683 V with *T* elevating from 25 to 175 °C, showing a positive temperature coefficient of 1 V/°C. As compared to a similarly-rated commercial SiC diode with a leakage current of ~10 μA under a reverse bias of 1200 V[28], the NiO/Ga₂O₃ HJD shows 10²-10³ times lower leakage current at high bias, reflecting the superior properties of UWBG materials.

The UIS setup and test procedure follows the industrial JEDEC standard[29], with the details elaborated in Supplementary Section S3. Figure 2b shows the UIS circuit schematic and the prototyped test board. In the UIS test, the switch (a power transistor with *BV* higher than that of the HJD) is first turned ON to charge the inductor ($L_{UIS}$); the switch is then turned OFF, forcing the HJD to withstand the surge energy stored in $L_{UIS}$. Figure 2c shows the test waveforms at *T* from 25 to 175 °C, revealing the desired avalanche waveforms: the HJD voltage clamps at $BV_{AVA}$ with the decreasing of $I_{AVA}$ from 30 A to zero, and the energy stored in $L_{UIS}$ is fully dissipated in the HJD in this 20 μs avalanche process. Repeating this avalanche process, Fig. 2c presents the sustainability of the HJD under 1-million cycles of repetitive avalanche tests. The device forward and reverse I–V characteristics before and after this cycle test show minimal parametric shifts, as presented in Supplementary Section S5. In addition, the $BV_{AVA}$ extracted from the

UIS waveforms shows a temperature coefficient identical to that extracted from the *I–V* characteristics (Fig. 2d). Note the $BV_{AVA}$ in the UIS test (1740 V) is higher than that in the *I–V* characteristics, due to a higher junction temperature in the UIS test under high $I_{AVA}$.

The UIS tests are then performed under various $L_{UIS}$ and the relevant charging time, both of which can alter $I_{AVA}$ and $E_{AVA}$. Avalanche waveforms are obtained under all these conditions. Figure 2a shows the expanded *I–V* characteristics that combine the quasi-static *I–V* curves (at low-current levels) and the $I_{AVA}$ - $BV_{AVA}$ data obtained from the UIS tests. Both the HJD and the reference SiC diode show a smooth transition between the two sets of data, suggesting a consistent avalanche across the main p-n junction under a wide range of $I_{AVA}$ (inconsistent avalanche locations would lead to abrupt transition[27]). Figure 2e shows the $I_{AVA}$ and $E_{AVA}$ under three $L_{UIS}$, revealing an $I_{AVA}$ up to 50 A and $E_{AVA}$ up to 730 mJ.

Critical band structure and carrier dynamics are scrutinized to understand the robust avalanche in the HJD. As shown in Fig. 2f, after I. I. is initiated in n-Ga₂O₃, the high *E*-field sweeps the produced electrons and holes to the cathode and the heterojunction, respectively. Under reverse bias, the staggered band produces no barriers for hole transport, which enables efficient hole removal (and high $I_{AVA}$). From the expanded avalanche *I–V* characteristics (Fig. 2a), the I. I. coefficients of electrons ($\alpha_n$) and holes ($\alpha_p$) in Ga₂O₃ are calculated. Note that only a theoretical $\alpha_n$[30] and no $\alpha_p$ have been reported for Ga₂O₃ previously. The calculated $\alpha_n$ and $\alpha_p$ are then fed into the TCAD simulation based on the Selberherr I. I. model[31]. The calculation and simulation models are detailed in Supplementary Section S4. Figure 2g shows the

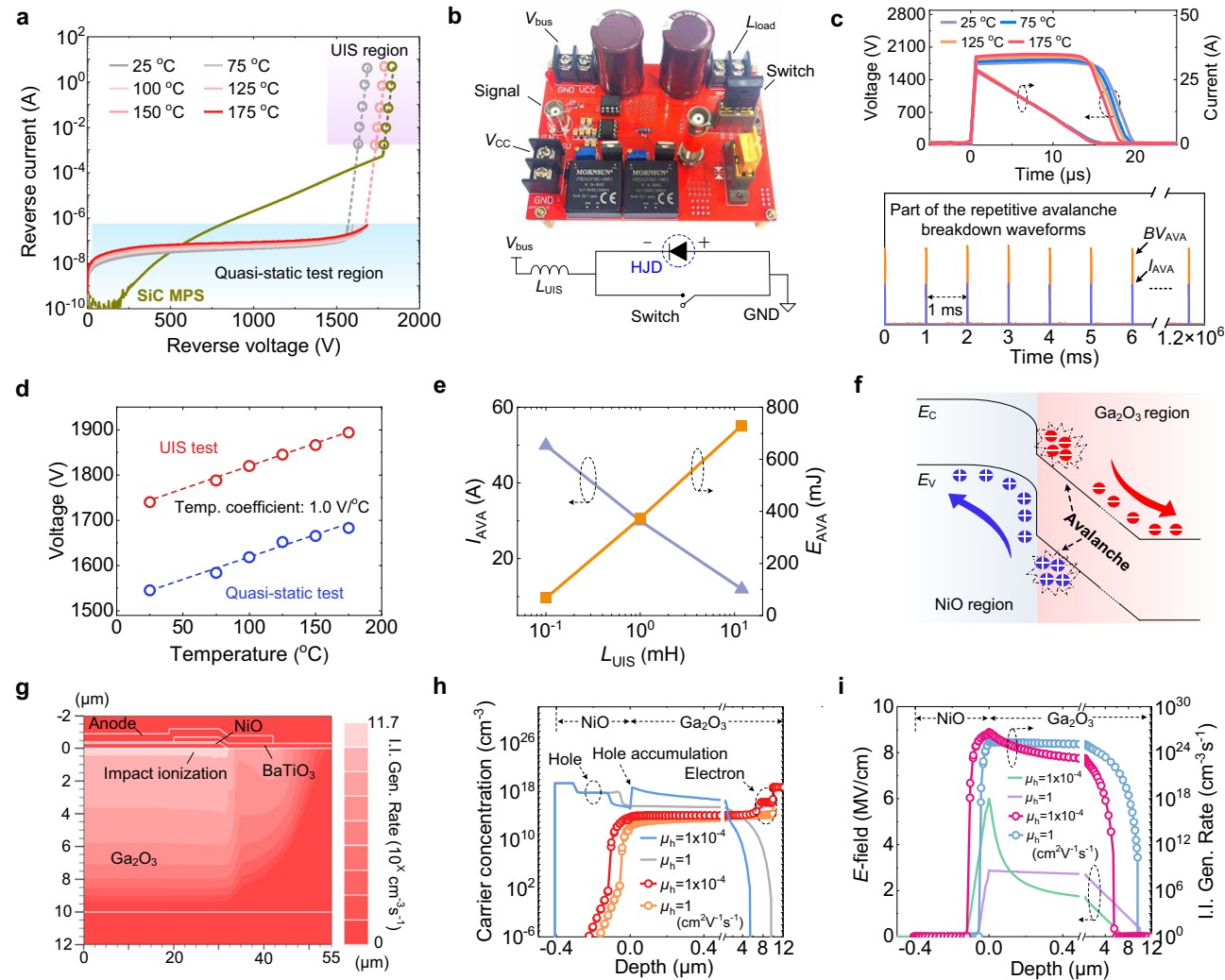

**Fig. 2 | Robust avalanche breakdown and the relevant carrier dynamics.**
**a** Temperature-dependent reverse $I$–$V$ characteristics of the HJD. **b** Photograph of
the UIS test setup and its circuit diagram. **c** Typical temperature-dependent UIS
voltage and current waveforms for the Ga₂O₃ HJD at a $L_{UIS}$ of 1 mH. **d** Temperature
coefficient values (1.0 V/°C) for $BV$ extracted from quasi-static and UIS

measurements, respectively. **e** $I_{AVA}$ and avalanche energy ($E_{AVA}$) as a function of $L_{UIS}$.
**f** Illustration of the carrier transport dynamics under the avalanche condition.
**g** Simulated contour of the impact ionization (I. I.) generation rate at $BV_{AVA}$.
**h, i** Simulated profiles of the electron and hole concentration, $E$-field, and I. I.
generation rate in the HJD under two different hole mobilities, at an $I_{AVA}$ of 30 A.

simulated contour of the I. I. generation rate at $BV_{AVA}$. The peak I. I. is
located in Ga₂O₃ near the junction and uniformly distributed in the
lateral dimension, confirming I. I. initiation in Ga₂O₃ and avalanche
process across the entire device active region.

The demonstration of avalanche in Ga₂O₃, which has not been
reported previously, also sheds light on the minority carrier (hole)
transport in Ga₂O₃, which remains controversial[32,33], and largely
unexplored under high $E$-field. As high $I_{AVA}$ hinges on efficient hole
removal, the I. I.-produced holes in Ga₂O₃ are believed to be exempt
from self-trapping under high $E$-field and drift with considerable
mobility ($\mu_p$). To estimate $\mu_p$, we simulate the avalanche dynamics with
$\mu_p$ of $10^{-4}$ and $1\,cm^2\,V^{-1}s^{-1}$ according to the low and high values theo-
retically predicted in the literature[32,34], ($\mu_p$ is the low-field hole mobi-
lity, and field-dependent mobility model is detailed in Supplementary
Section S4). Note that higher $\mu_p$ values, e.g., ~1.2 $cm^2/V\,s$[34] and
8-10 $cm^2/V\,s$[22,35], have been reported experimentally. Here we use two
lower $\mu_p$ in the simulation mainly to consider the worst scenario
of avalanche, as a high $\mu_p$ can allow for a more efficient hole removal
and thus supports a high avalanche current. At $BV_{AVA}$, the drift velocity
does not reach saturation under either $\mu_p$ values. Figure 2h, i shows the
simulated contours of carrier concentrations, $E$-field and I. I.

generation rate for the two $\mu_p$, respectively. The low $\mu_p$ would induce
serious hole accumulation and high $E$-field crowding at the hetero-
junction, making it unlikely to sustain a stable avalanche. These effects
are eliminated for $\mu_p = 1\,cm^2\,V^{-1}s^{-1}$, rendering it a more reasonable $\mu_p$ to
explain the avalanche in n-Ga₂O₃.

## Surge current and reverse recovery characteristics

While avalanche represents the HJD's robustness at reverse bias, surge
current measures its capability to withstand forward overcurrent.
Here, a 10-ms half-sinusoidal current pulse with an adjustable ampli-
tude is employed for surge current characterization following the
JEDEC standard[36]. Figure 3a shows the circuit schematic and the pro-
totyped setup, with the circuit design detailed in Methods. Figure 3b, c
shows the current and voltage waveforms in the surge current tests
with increased amplitude. The HJD can withstand over 50 A surge
current, under which condition the forward voltage approaches 14 V.
Based on these time-resolved data, the surge current $I$–$V$ locus are
plotted in Fig. 3d for the HJD and a reference Ga₂O₃ Schottky barrier
diode (SBD) fabricated on the same wafer. The surge current and
voltage characteristics of the reference Ga₂O₃ SBD are shown in Sup-
plementary Section S6. The locus of the HJD and SBD shows an

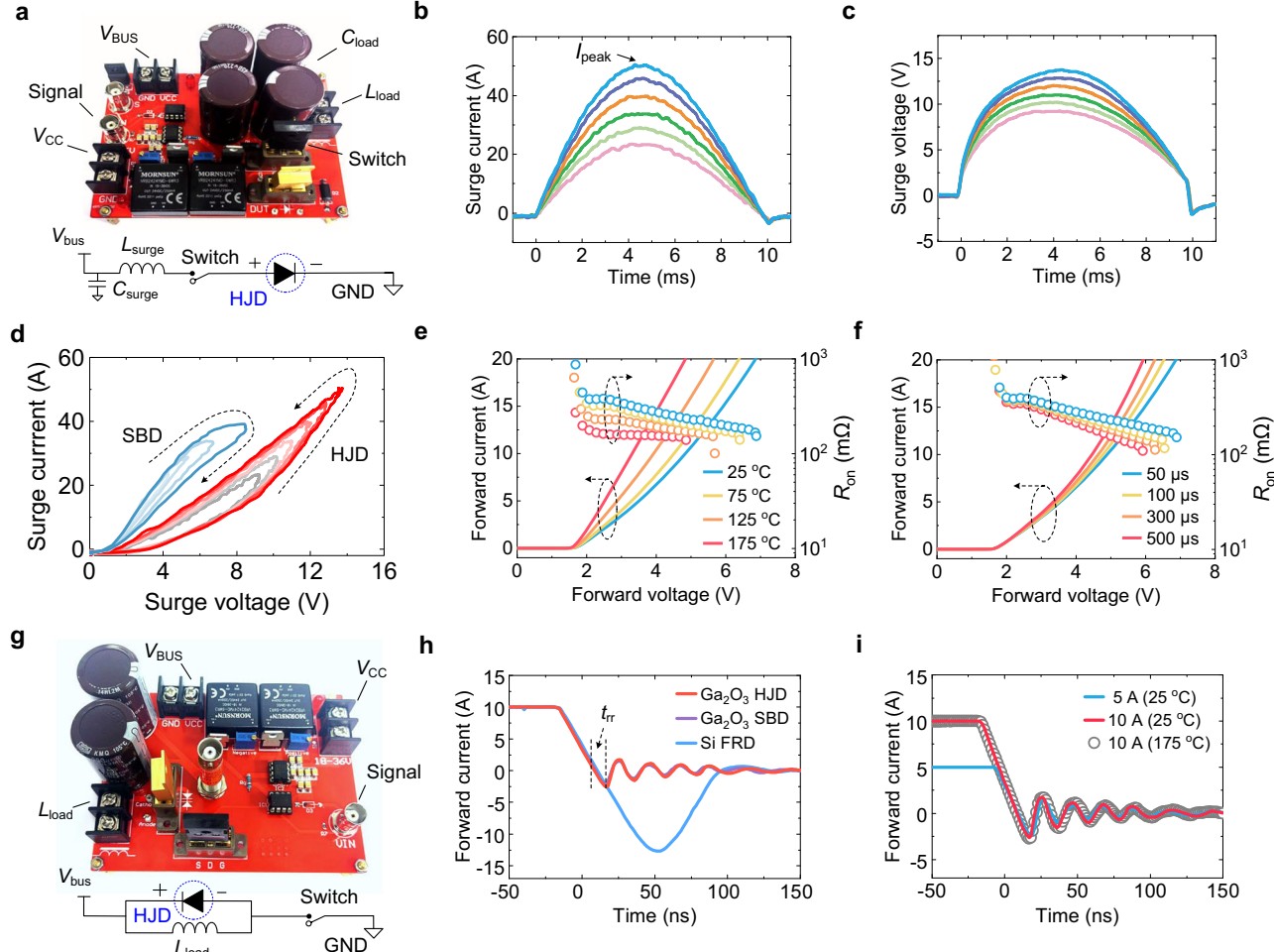

**Fig. 3 | Surge current and reverse recovery characteristics. a** Photograph of the surge current test setup and its circuit diagram. **b**, **c** Surge current and voltage waveforms of the HJD, respectively. **d** Surge $I$–$V$ locus of the HJD and the reference $Ga_2O_3$ SBD, each set of locus with various amplitudes. **e**, **f** Quasi-static temperature-dependent $I$–$V$ characteristics and pulse $I$–$V$ characteristics of the HJD, respectively.

**g** Photograph of the reverse recovery test setup and its circuit diagram. **h** reverse recovery characteristics of the $Ga_2O_3$ HJD, the reference $Ga_2O_3$ SBD, and a commercial Si fast-recovery diode (FRD). **i** reverse recovery waveforms of the HJD with different conditions.

anticlockwise and clockwise signature, respectively, which signifies negative and positive temperature coefficients ($\eta_T$) of the differential on-resistance ($R_{ON}$)[12,37]. The $R_{ON}$'s negative $\eta_T$ in the HJD is further confirmed by the forward $I$–$V$ characteristics measured in the DC mode at elevated $T$ (Fig. 3e) and in the pulse mode with various pulse widths (Fig. 3f). The differential $R_{ON}$ decreases with the increased pulse width and forward voltage.

While a positive $\eta_T$ of $R_{ON}$ is expected for the unipolar SBD due to the mobility drop at high $T$, the negative $\eta_T$ suggests bipolar conductivity modulation in the HJD. Bipolar conduction in homo-junction power devices is usually dominated by the high-level carrier injection into the lightly-doped drift region, e.g., hole injection into n⁻-type layer, which typically lead to the considerable reverse recovery[38]. When bipolar devices fast switches from a high forward current to high reverse bias, a wide depletion region has to be established in the lightly-doped drift region, requiring the removal of minority carriers that previously fill this region. This leads to a usual trade-off between surge robustness and switching speed for bipolar power devices.

To explore if a similar trade-off holds for the NiO/$Ga_2O_3$ hetero-junction, reverse recovery characterization is performed for the HJD, the reference $Ga_2O_3$ SBD, and a similarly-rated commercial Si fast-recovery diode (FRD)[39], with the circuit schematic and prototype shown in Fig. 3g and detailed in Methods. As shown in Fig. 3h, i, the 1200-V reverse

recovery of the HJD is similar to the unipolar $Ga_2O_3$ SBD and much faster than the bipolar Si FRD. Moreover, the reverse recovery waveforms are nearly independent of the forward current (5–10 A) and temperature (25–175 °C), suggesting the reverse recovery waveform is dominated by the capacitive ringing instead of minority carrier recombination. The recombination process, if any, should be faster than the duration of the first ringing. Based on the method in ref. 12 and the reverse recovery time ($t_{rr}$) of 12.79 ns extracted from the first ringing waveform, the maximum hole lifetime in $Ga_2O_3$ is estimated to be 6.50 ns.

## Microscopic EBIC characterization

The high surge current capability albeit minimal reverse recovery in the HJD suggests a different origin of conductivity modulation as compared to conventional bipolar devices. We suspect the origin to be the electron injection into NiO. Under low forward bias, trap-assisted tunneling dominates the interfacial recombination, while under the elevated forward bias, high-level minority carrier (electrons) injection into the NiO layer occurs via tunneling or thermionic emission across the interfacial barrier at the conduction band. Injected electrons diffuse in the neutral region of NiO and contribute to current conduction, reducing the whole $R_{ON}$ of the HJD[40]. Under reverse bias, as the depletion mainly occurs in the lightly-doped $Ga_2O_3$, minimal minority carries need to be recombined in p-NiO for switching to occur. Hence, the carrier dynamics in NiO insignificantly impact the device reverse recovery.

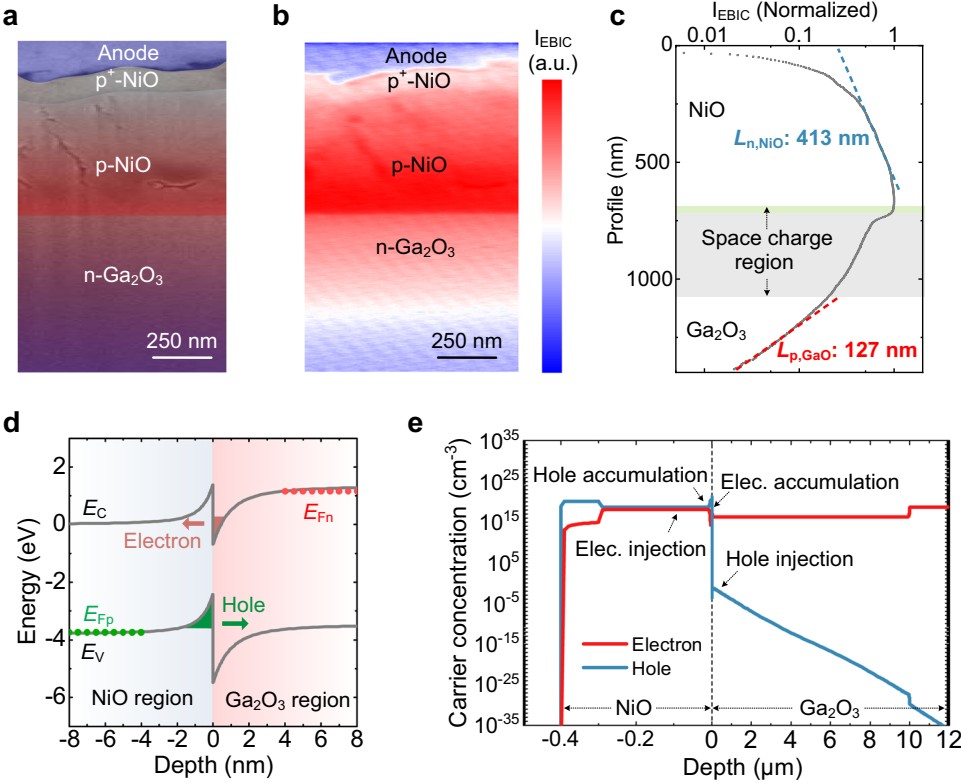

**Fig. 4 | EBIC characterization and the carrier dynamics under the surge condition. a** False color cross-sectional scanning electron microscopy (SEM) image of the HJD. **b** The corresponding colored EBIC map of the heterojunction region under zero bias and at room temperature. **c** EBIC profile extracted by the integration of EBIC intensity map in the vertical direction. **d** Illustration of the carrier transport dynamics under the high forward current. **e** Simulated distribution of minority carriers on both sides of the HJD at a forward voltage of 6 V.

To pursue the microscopic visualization of carrier diffusion, we apply microscopic EBIC characterization, an SEM-based technique widely used to image carrier dynamics to the NiO/Ga$_2$O$_3$ junction[41]. The EBIC signal is recorded simultaneously with the SEM detector signal, thereby allowing us to spatially correlate the EBIC signal. Figure 4a shows the cross-sectional SEM image of the HJD, and the corresponding EBIC map is provided in Fig. 4b. In the presence of built-in *E*-field in the depletion region, the EBIC signal decays from the junction interface to both sides of the junction, where a higher current signal is observed in NiO. This indicates more minority carriers are injected into NiO, since only minority carriers contribute to the induced current[42]. The exponential decay of the EBIC signals is well described in terms of $I = I_0 e^{-x/L}$, where $I_0$ is the maximum intensity and $x$ is the distance from the junction[41]. Accordingly, the minority carrier diffusion length ($L$) for electrons in p-NiO and holes in n-Ga$_2$O$_3$ are determined to be 413 and 127 nm, respectively (Fig. 4c). The difference to the reported values in[43,44], could be result of the variation in epitaxial quality, synthesis methods, and material conductivities. Consequently, the minority carrier lifetime ($\tau$) for electrons in p-NiO and holes in Ga$_2$O$_3$ are extracted to be 124.0 and 6.2 ns, respectively, based on the relation $L = (D \times \tau)^{1/2}$ and the carrier diffusion coefficient ($D$) in respective materials[45]. The hole lifetime in Ga$_2$O$_3$ is consistent with the maximum value estimated from the reverse recovery waveform. In addition, in terms of Einstein's relation, the hole mobility in Ga$_2$O$_3$ is estimated to be 1 cm$^2$ V$^{-1}$s$^{-1}$, which is consistent with the value estimated from the avalanche simulations shown in Fig. 2h, i.

The EBIC results reveal that the $L$, $\tau$, and minority carrier quantity in Ga$_2$O$_3$ are much smaller than those in NiO, making the conductivity modulation occurs predominantly in NiO. Accordingly, the schematic of band structure and carrier dynamics in the HJD at high forward bias is illustrated in Fig. 4d. At the heterojunction, electrons and holes primarily tunnel through the barriers produced by band offsets, whilst undergoing a trap-assisted band-to-band recombination[46]. Feeding the parameters extracted from EBIC into the simulation, the contours of electrons and holes are shown in Fig. 4e at a forward voltage of 6 V. High concentration of electrons approaching 10$^{20}$ cm$^{-3}$ are present in the entire NiO region, while the hole concentration in Ga$_2$O$_3$ is much lower and drops rapidly away from the junction. The simulation results validate the more pronounced minority carrier injection in NiO under the surge current condition.

## Discussion

The intriguing physics of the NiO/Ga$_2$O$_3$ heterojunction enables the fabricated HJD to deliver a breakthrough trade-off between device performance and robustness. Achieving low $R_{ON}$, high current, and high $BV$ concurrently is a major pursuit for power device technologies[1]. As shown in Fig. 5a, the $R_{ON}$ and $BV$ trade-off of our HJD is the record-breaking among the reported ampere-class Ga$_2$O$_3$ diodes with forward current >1 A[37,47–55], In addition, the benchmark of specific on-resistance ($R_{on,sp}$) versus BV of the ampere-class Ga$_2$O$_3$ HJD against small-area Ga$_2$O$_3$ power diodes is shown in Supplementary Section S9. Figure 5b benchmarks the $E_{AVA}$ density of our HJD with homojunction-based Si, SiC, and GaN devices with a similar $BV_{AVA}$. The $E_{AVA}$ density of our HJD is significantly higher than that of Si devices and comparable to the highest values reported in SiC and GaN devices[12,27,28,56–62], In addition, as shown in Fig. 5c, our HJD shows an excellent combination of high surge current and surge energy comparable to the highest values reported in Si, SiC, and GaN devices[12,39,60,61,63–67], Finally, as shown in Fig. 5d, our HJD breaks the trade-off between the reverse recovery time and surge energy in Si, SiC, and GaN devices[12,28,39,61,63–67], enabling a much lower switching speed and loss, whilst demonstrating the state-of-the-art robustness.

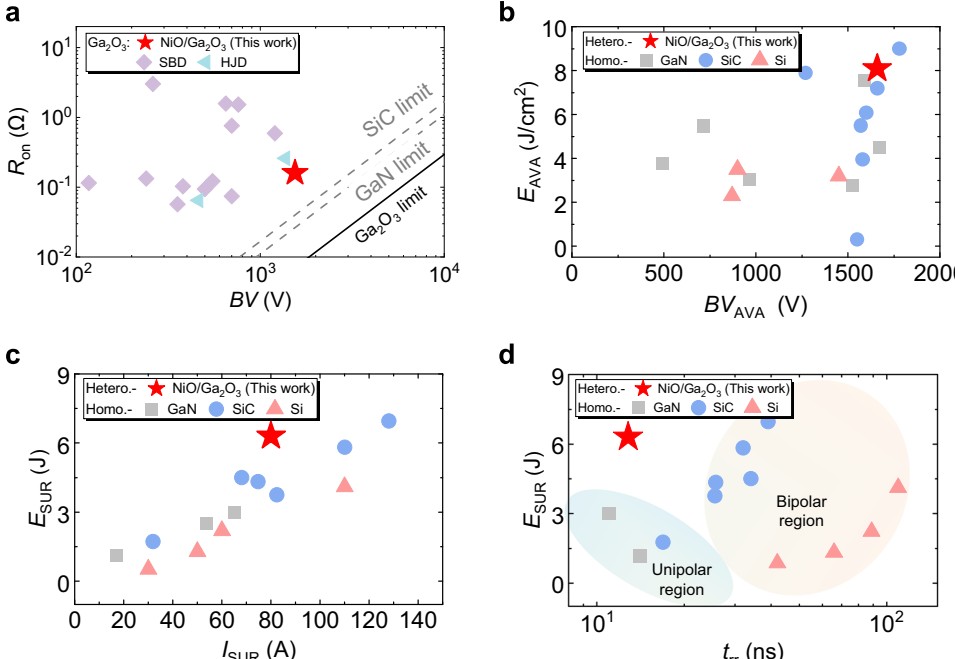

**Fig. 5 | Benchmarking performance and robustness of the Ga$_2$O$_3$ HJD against commercial Si, SiC, and GaN power diodes. a** $R_{on}$ versus $BV$ benchmark for all the ampere-class Ga$_2$O$_3$ power diodes. **b** $E_{AVA}$ versus $BV_{AVA}$ benchmark, **c** Surge energy ($E_{SUR}$) versus peak surge current ($I_{SUR}$) benchmark, and **d** $E_{SUR}$ versus reverse recovery time ($t_{rr}$) benchmark, all for Ga$_2$O$_3$, GaN, SiC, and Si power diodes.

In summary, this work demonstrates a NiO/Ga$_2$O$_3$ heterojunction architecture with high avalanche and surge current robustness, minimal reverse recovery, as well as an excellent trade-off between $R_{ON}$ and $BV$. The robustness of the large-area, packaged HJD is characterized by in-circuit tests following industrial standards, revealing the survival of >1700 V overvoltage and >50 A overcurrent. This robustness is attributable to the effective edge termination design, high-quality heterojunction, staggered band alignment, and conductivity modulation. Benefitting from the asymmetric minority carrier lifetimes, conductivity modulation is dominant in p-NiO, which is visualized through microscopic EBIC characterization and enables minimal reverse recovery in switching applications. The avalanche and surge current characteristics also provide physical insights into the fundamentals of hole transport dynamics in Ga$_2$O$_3$, particularly the parameters including impact ionization coefficient, high-field mobility, and a lifetime of minority carriers. This heterojunction is promising for making robust power devices in semiconductors lacking bipolar doping for applications in electric vehicles, aerospace, grid, and renewable energy processing.

## Methods
### Epitaxial structure
The epitaxial structure was grown by hydride vapor phase epitaxy (HVPE) on a conductive Sn-doped (001) $\beta$-Ga$_2$O$_3$ substrate, consisting of a 10-μm Si-doped $\beta$-Ga$_2$O$_3$ drift layer with an electron concentration of $1.7 \times 10^{16}$ cm$^{-3}$. Based on the Hall measurements of the controlled NiO samples on semi-insulating substrates, hole mobilities corresponding to p-NiO ($5.8 \times 10^{17}$ cm$^{-3}$) and p$^+$-NiO ($2.9 \times 10^{19}$ cm$^{-3}$) layers were determined to be 0.87 cm$^2$/V s and 0.34 cm$^2$/V s, respectively. Schematic of the processing steps is shown in Supplementary Section S1. The current-voltage characteristics of the p-NiO layer are shown in Supplementary Section S8.

### Device fabrication
The device fabrication started with the substrate thinning from 640 μm to 150 μm by chemical mechanical polishing process, followed by substrate cleaning via ultrasonic treatment in acetone and soaking.

Then, the Ga$_2$O$_3$ epi-wafers were annealed at 500 °C for 5 hours under the O$_2$ ambient to partially compensate the donors in the epi-layer. The back-side Ohmic contact (cathode) was formed by the Ti/Au (20/80 nm) deposition through electron beam evaporation (EBE), followed by rapid thermal annealing at 500 °C for 1 min under N$_2$ ambient. This annealing process has been reported to be able to effectively passivate near-surface defects in Ga$_2$O$_3$ epi-layer[68]. Note that the inter-diffusion of Sn from substrate and Si from epi-wafer is expected to be negligible at this annealing temperature[69,70]. Subsequently, by using an angled shadow mask, a 400-nm-thick double-layered NiO film with adjustable bevel angle was deposited on the Ga$_2$O$_3$ drift layer by RF magnetron sputtering technique at room temperature. During the sputtering process, the substrate was rotated at 4 rpm to enhance the film uniformity. The target was high-purity (99.99%) NiO ceramics. To alleviate the damage induced by sputtering plasma, the initial RF power was 50 W, and then increased to 150 W when the NiO thickness was above 20 nm. The distance between the target and the wafer was maintained at 13 cm. The growth pressure was 0.6 Pa in an Ar/O$_2$ mixed ambient, and the flux ratios of Ar/O$_2$ were tuned from 20:1 to 2:1 to modulate hole concentration in the double-layered NiO. A 300-nm-thick BaTiO$_3$ was also deposited by RF magnetron sputtering at the same growth pressure of 0.6 Pa in an Ar/O$_2$ mixed ambient with a flux ratio of 10:1, followed by annealing at 300 °C in oxygen ambient for 1 hour. The contact window was opened by a lift-off process, and Ni/Au (300/200 nm) metal stack was deposited by EBE to form the anode contact, producing an active area of $3 \times 3$ mm$^2$.

### Device package
The device was sealed in a TO-220 package for circuit testing. For the comparison of surge current characteristics, 9-mm$^2$ Ni/$\beta$-Ga$_2$O$_3$ SBDs were also fabricated on the same wafer by identical processes except for the p-NiO deposition underneath the Schottky anode.

### Device static electrical characterizations
The quasi-static forward/reverse $I$–$V$ characterizations were performed by a B1505 power device analyzer. The pulse measurement mode in the

B1505 analyzer was adopted to characterize the forward current above 1 A (Fig. 3e, f), while the high-resolution DC mode was used to measure the log-scale forward $I-V$ characteristics of the HJD (Supplementary Section S7). The $C-V$ characteristics were measured by using an E4980A precision LCR meter at room temperature.

## Device circuit-level characterizations

The UIS, surge current, and reverse recovery characterizations were carried out by a customized circuit test platform. All test circuits and methods were formulated according to the Joint Electron Device Engineering Council (JEDEC) standards. Photographs of test circuits and experimental platforms are presented in Supplementary Section S3.

## EBIC and TEM characterizations

EBIC measurement was carried out in an FEI Helios 600 NanoLab Dual-beam FIB system equipped with Kleindiek Nano Control NC40 nano-manipulators and low-current measurement units. The top- and bottom electrodes were contacted to the nanomanipulator and sample stage, respectively, allowing electrical current to flow and pass through a current amplifier. The electron beam direction is perpendicular to the surface of the junction, and the acceleration voltage and electron beam current were 15 kV and 0.17 nA, respectively. Line-scan measurement from the top electrode down to the $Ga_2O_3$ drift layer was performed to extract the profile of the EBIC current across the junction. Scanning TEM, bright-field high-resolution TEM, and energy dispersive X-ray (EDX) spectroscopy elemental mapping were performed using an FEI Tecnai F-20 microscope (FEI TF-20), operated at an acceleration voltage of 200 kV.

## Simulations

The Technology Computer-Aided Design (TCAD) device simulations were performed using the Silvaco TCAD device package. Additional details for the simulations are provided in Supplementary Section S4.

## Data availability

The authors declare that the data supporting the findings of this study are available within the paper and its supplementary information files. Source data are provided with this paper.

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

## Acknowledgements

We thank Dr. Yu Deng for his assistance in TEM and EDX measurements, and Silvaco Inc. for collaboration and support of material and device TCAD simulations. This work was supported in part by the National Key R&D Program of China (No. 2022YFB3605403), the National Natural Science Foundation of China (62234007, 62293521, U21A20503, and U21A2071), the Key-Area R&D Program of Guangdong Province (2020B010174002). Access to the EBIC facility is made possible through support from the Australian National Fabrication Facility, ACT Node. We also thank Han Wang at the University of Southern California for valuable feedback on the manuscript.

## Author contributions

J.D.Y., Y.H.Z., R.Z., and H.L. proposed and directed the research. H.H.G. performed device design and fabrication. F.Z. and H.H.G. performed the characterization and data analysis. M.X. and Y.W.M. performed the

physical-based simulations and theoretical calculations. Z.P.W. and X.X.Y. performed the Hall and TEM characterizations. L.L. and L.F. performed the EBIC characterization and analysis. Y.Y., F.F.R., S.L.G. H. H. T., and Y.D.Z discussed the results and commented on the manuscript. Y.H.Z., J.D.Y., and F.Z. wrote the manuscript, and all authors approved the final version of the manuscript.

## Competing interests

The authors declare no competing interests.
