## [Peer Review File · Nature Communications]

An avalanche-and-surge robust ultrawide-bandgap heterojunction for power electronicsREVIEWER COMMENTS

Reviewer #1 (Remarks to the Author):

This article provides a validation of avalanche and surge robustness in NiOx/Ga2O3 Bipolar diodes for power electronic applications. Surge current robustness has been validated by several authors previously, but this paper also validates the avalanche mechanism - this can be considered as the novelty of the manuscript. The authors also provide possible physical explanations with energy-band profiles and electric field distributions.

Here are my review comments:

1. Since the diode is nearly a P+N abrupt diode (doping in p-NiOx is $5.8 \times 10^{17} \text{ cm}^{-3}$, compared to $1.7 \times 10^{16} \text{ cm}^{-3}$ in n-Ga2O3), the depletion region is mostly toward the Ga2O3 side. Effective equilibrium depletion region is $\sim 250 \text{ nm}$, as seen in Fig. 1d. Can the authors comment on the effective depletion width at 1600 V reverse bias. Kindly provide the CB-VB profile near the NiOx/Ga2O3 interface (zoomed image) at 1600 V reverse bias.
2. The authors claim that I.I process initiates at 1500 V, Whereas the previous reports validated that NiOx/Ga2O3 bipolar diodes went into breakdown at $>4.5 \text{ kV}$ (for $7.5 \mu\text{m}$ drift layer) and $>8.3 \text{ kV}$ at $13 \mu\text{m}$ drift layer [Ref. 13]. Kindly clarify the possible reason for attaining breakdown at an electric field ($\sim 4.6 \text{ MV/cm}$) which is much below the critical electric field of Ga2O3.
3. Several NiOx/Ga2O3 bipolar HJD's have been reported so far, whereas Fig. 5a only mentions a couple of data points (I believe the reason being Ampere-class). However, I think the overall contact area eventually determines the current levels, and hence I feel other reports also deserve a spot in Fig. 5a.
[Kindly see the following papers: 10.1109/TPEL.2021.3123940, 10.1109/TED.2021.3091548].
4. The authors claim hole injection into the Ga2O3 layer during forward bias (Fig. 4e). However, it is not clear how holes are able to overcome the large VB barrier (authors claim a 3.2 eV VB-offset between NiO and Ga2O3). Kindly explain.
5. Kindly provide the barrier height and depletion width information in Fig. 4d. If the hole injection is tunneling assisted, kindly provide the tunneling probability of holes in overcoming the VB barrier potential during forward bias.
6. The authors claim "textbook-like avalanche" for Fig. 2a. However, the textbook model also talks about reverse saturation current increasing by 2x in every 10 degree rise in temperature. The net increase in the reverse current is less than 2X when temperature is increased from 25 C to 175 C. I am wondering what may be the cause of this increase, considering the bandgap may be high enough to prevent thermal generation. Kindly provide a justification for this in the manuscript so that the general audience is aware of the overall scenario.

Reviewer #2 (Remarks to the Author):

1. Key results:

The present work is related to very hot topic related to the new generation ultra wide band gap material - Ga2O3 potential application into power electronics. Authors present the study of avalanche and surge robustness study under extremely high electric field and current density; this knowledge is very important for development power devices based on Gallium Oxide. The main achievement is:

- Give insights into the fundamentals of hole transport dynamics in Ga2O3
- Estimation experimentally impact ionization coefficients
- The results open a new pathway for making robust Ga2O3/NiO heterojunction-based power devices

2. Validity: Does the manuscript have flaws which should prohibit its publication? If so, please provide details. No

3. Originality and significance:

The majority of presented results are of immediate interest to people working on wide and ultra wide band gap materials fundamental properties as well for people working in the field of power electronics.

Except: Author should not state as first original demonstration/measurements of hole carrier diffusion length and life time determination. Such values were reported in Appl. Phys. Lett. 112, 082104 (2018); doi: 10.1063/1.5011971; 10.1149/2.0101702jss; APL Materials 10, 031106 (2022); <https://doi.org/10.1063/5.0086449>

4. Data & methodology:

1) there are lack of information/ precisions in the part related to device fabrication Process S1:

- what is the origin (how are they fabricated?) of conductive Sn-doped (001) β -Ga₂O₃ substrates? What is the Sn doping level and electrical resistivity?

- What was the growth temperature of Si:Ga₂O₃ epilayers? What is the Si doping level (exact concentration) in Ga₂O₃ epi-wafers?

- How $n = 1.7 \times 10^{16} \text{ cm}^{-3}$ electron concentration for epilayers was measured? And what is the mobility value?

- regarding Annealing a) If the goal of annealing as it is stated was to decrease donor concentrations in Si:Ga₂O₃ to have resistive layer why epilayers were doped during the growth? It is not written what is the optimal electron concentration targeted by authors for this device structure.

b) annealing at 500 °C for 5 hours under the O₂ ambient: can occur the interdiffusion of Sn from substrate and Si from epi-wafer? How changes (if this is a case) interface profile after annealing? It is not addressed this issue by authors.

- What was the electron concentration value after annealing? Can Ga₂O₃:Si epilayer be still considered as an n type or it becomes semi-insulating?

-

- It is not mentioned what is I-V characteristic of Ti/Au (20/80 nm) metal contacts deposited back side on substrate?

- It is not discussed, how were measured hole concentrations in NiO layers? What is hole mobilities?

- Authors do not comment obtained values of impact ionization coefficients for holes and electrons. Why I.I. coefficient is higher for holes? How it correlates with low hole mobilities and higher (than for electrons) effective mass? Will be good to compare with impact ionization coefficients for SiC measured experimentally.

5. Conclusions: are the conclusions and data interpretation robust, valid and reliable?

I think so.

6. Suggested improvements: please list additional experiments or data that could help strengthening the work in a revision.

- Page 4 Ref. 23 will be good to give corresponding values for SiC for comparison.

- Page 5 regarding simulation where input is μ values. Authors could take values for hole mobilities which are experimentally measured in undoped Ga₂O₃ thin films from for example: Journal of Materials Chemistry C, 2019, 7, 10231 <https://doi.org/10.1039/C9TC02910A>; Materials Today Physics, 3, 118 (2017) <https://doi.org/10.1016/j.mtphys.2017.10.002> ($\mu = 8-10 \text{ cm}^2/\text{Vs}$) value could better explain perhaps avalanche in n-Ga₂O₃.

- In the model for avalanche simulation, Table S2: Will be more interesting and realistic to use data from experiment, i.e. measured values of mobilities.

- supplementary section S1: in all figures, substrate and epilayer doping will be good to indicate: i.e. Si:Ga₂O₃ and Sn:Ga₂O₃, indeed both of them are n-type.

7. References: does this manuscript reference previous literature appropriately?

It is not very clear the connection of the present work with following publications:

*b-Ga2O3 hetero-junction barrier Schottky diode with reverse leakage current modulation and BV2/Ron,sp value of 0.93 GW/cm2 Appl. Phys. Lett. 118, 122102 (2021); doi: 10.1063/5.0044130
** Zhang, J. et al. Ultra-wide bandgap semiconductor Ga2O3 power diodes. Nat. Commun. 13, 3900(2022)

It should be corrected cited those articles and underlined the similarity and the difference with the results in present manuscript.

~~~~~

- Hsiao-Hsuan Wan, J. Vac. Sci. Technol. A 41, 032701 (2023) is very important to cite.

In Page 2: Authors state idea that p-type doping has been unsuccessful. That is wrong, since there are reports demonstrating native hole conductivity and Zn and N doping related hole conductivity as well.

Journal of Vacuum Science & Technology A 40, 043401, 2022.

<https://doi.org/10.1116/6.0001766>; Journal of Materials Chemistry C, 2019, 7, 10231

<https://doi.org/10.1039/C9TC02910A>

<https://doi.org/10.1016/j.mtphys.2021.100356>

Though and evidently, reported hole concentrations are not enough for device functioning.

~~~~~

It is interesting to compare carrier diffusion length for Ga2O3 reported in Applied Materials letters, APL 10, 031106 (2022); <https://doi.org/10.1063/5.0086449>

-Minority (holes) diffusion length and lifetimes for Ga2O3 have been already reported by authors in: Appl. Phys. Lett. 112, 082104 (2018); doi: 10.1063/1.5011971

Authors should cite this publication and compare their results.

~~~~~

It was reported reported experimentally estimated break down electrical field for Ga2O3 in the <https://doi.org/10.1016/j.mtphys.2020.100263>, will be interesting if authors will make comment regarding ...

8.Clarity and context: Is the abstract clear, accessible? Are abstract, introduction and conclusions appropriate? Yes

9.Please indicate any particular part of the manuscript, data, or analyses that you feel is outside the scope of your expertise, or that you were unable to assess fully.

I cannot assess properly the part related to "Surge current and Avalanche characterizations"

## Response Letter (Manuscript ID: NCOMMS-23-13639)

We would like to thank Reviewers for their constructive and helpful suggestions. In the *Response Letter*, all comments from reviewers have been addressed point-by-point, and the corresponding revisions have been highlighted in BLUE in the revised manuscript. The replies are listed below.

|               | I. Response to First reviewer's comments | II. Response to Second reviewer's comment |
|---------------|------------------------------------------|-------------------------------------------|
| Pages numbers | 1~10                                     | 11~19                                     |

### **I. Reply to the 1st reviewer's comments**

*This article provides a validation of avalanche and surge robustness in NiOx/Ga2O3 Bipolar diodes for power electronic applications. Surge current robustness has been validated by several authors previously, but this paper also validates the avalanche mechanism - this can be considered as the novelty of the manuscript. The authors also provide possible physical explanations with energy-band profiles and electric field distributions.*

**Comment (1):** *Since the diode is nearly a P+N abrupt diode (doping in p-NiOx is  $5.8e^{17} \text{ cm}^{-3}$ , compared to  $1.7e^{16} \text{ cm}^{-3}$  in n-Ga2O3), the depletion region is mostly toward the Ga2O3 side. Effective equilibrium depletion region is ~250 nm, as seen in Fig. 1d. Can the authors comment on the effective depletion width at 1600 V reverse bias. Kindly provide the CB-VB profile near the NiOx/Ga2O3 interface (zoomed image) at 1600 V reverse bias.*

**Reply:** Thanks for the enthusiastic comments and unanimous recognition of the innovation of avalanche robustness achieved in Ga2O3 HJD. On the other hand, we would like to note that, despite the prior report on the surge current values, to the authors' knowledge, this paper for the first time unveils the underlying physics for carrier transport to enable such a high surge current capability in the NiO/Ga2O3 HJD. The reason why avalanche and surge current tests are both included in this article is partly from the application perspective (as they represent the two most important robustness of power devices) but also from the physical perspective, as the carrier dynamics possess correlations under these two conditions and they share common correlations with the microscopic EBIC results. Hence, we believe that, in addition to the avalanche result, the surge current section, as well as its correlation with the avalanche section and EBIC results, all exhibit considerable innovation.

As the reviewer stated, the p-NiO/n-Ga2O3 HJD here inherently form a P+-N abrupt junction, where the effective depletion region is mostly toward on the Ga2O3 side at reverse bias. Based on the calibrated band simulations, the effective depletion region is about 250 nm under unbiased equilibrium condition; at a reverse bias of 1.24 kV, the 10-

$\mu\text{m}$  n-Ga2O3 drift layer is fully depleted. When the bias voltage continues to increase, the depletion region will slightly extend from the fully-depleted drift region ( $1.7 \times 10^{16} \text{ cm}^{-3}$ ) into the heavily doped substrate ( $6.0 \times 10^{18} \text{ cm}^{-3}$ ). Note that the depletion width in the substrate is very small due to the high doping concentration. Thus, at a reverse bias of 1.6 kV, the diode is operated under the punch-through condition, and the effective depletion width is nearly the same as the drift region thickness of about 10  $\mu\text{m}$  in this work. Correspondingly, the extracted band-depth profile near the NiO/Ga2O3 interface at 1.6-kV reverse bias is shown in Fig. R1.

Fig. R1. Energy band profile of the Ga2O3 HJD at a reverse bias of 1.6 kV.

**Comment (2):** The authors claim that I.I process initiates at 1500 V, Whereas the previous reports validated that NiOx/Ga2O3 bipolar diodes went into breakdown at  $>4.5 \text{ kV}$  (for 7.5  $\mu\text{m}$  drift layer) and  $>8.3 \text{ kV}$  at 13  $\mu\text{m}$  drift layer [Ref. 13\*]. Kindly clarify the possible reason for attaining breakdown at an electric field ( $\sim 4.6 \text{ MV/cm}$ ) which is much below the critical electric field of Ga2O3.

**Reply:** Thanks for the comment. As pointed out by the reviewer, the peak electric field ( $E_{\text{peak}} \sim 4.6 \text{ MV/cm}$ , see Fig. 1e in the revised manuscript) under breakdown transient in this work is indeed lower than the best result previously reported ( $> 6 \text{ MV/cm}$ ,  $8.3 \text{ kV}/13 \mu\text{m}$  drift layer) in Ref. [13\*] of the manuscript. To the best of our knowledge, this referenced work is among the only one or two Ga2O3 power devices published with ultra-high  $BV$  and ultra-high  $E$ -field, which is an extraordinary performance breakthrough but may be elusive. Note that the device reported in Ref. [13\*] is a small-area device, which has an anode radius of only 75  $\mu\text{m}$  with a corresponding electrode area 500 times smaller than that of our device (ampere-class).

A more meaningful reference of the state-of-the-art for our device could be the large-area (ampere-class) Ga2O3 devices. We believe this is a more fair comparison for two following reasons.

(1) Industrial power devices for practical power (1) applications usually require a minimal current rating of at least

several amperes (more common is a current rating of tens of amperes even up to hundreds of amperes), so the ampere-class device performance represents the real state-of-the-art of a power device technology;

(2) For  $\text{Ga}_2\text{O}_3$ , an emerging semiconductor technology that is still in the early stage of development, there is a large gap between the performance of small- and large-area devices, due to the non-uniformity in fabrication process and wafer material properties. Larger device regions are usually more susceptible to non-uniform doping, non-uniform dislocation distribution [1], metal/semiconductor contact spiking, and local  $E$ -field crowding [2]. In particular, breakdown voltage is usually determined by the spot with the ‘weakest’ material property (e.g., highest defect density or doping). Consequently, it is much more challenging to achieve high  $E$ -fields in large-area devices.

Fig. R2 depicts the  $E_{\text{peak}}$  versus  $BV$  for reported large-area  $\text{Ga}_2\text{O}_3$  rectifiers (ampere-class), including SBDs, HJDs and junction barrier Schottky diode (JBSS) [2-16]. Overall, the  $E_{\text{peak}}$  of most reported  $\text{Ga}_2\text{O}_3$  diodes is below 3 MV/cm, which is even lower than the critical  $E$ -field of GaN and SiC materials. **The  $E_{\text{max}}$  of about 4.6 MV/cm achieved in this work highlights the highest  $E_{\text{peak}}$  reported in ampere-class  $\text{Ga}_2\text{O}_3$  devices. We attribute this achievement to the effective edge termination that incorporates the small-angle beveled junction termination extension (JTE) and high- $k$  field plate.**

Another point that this work differs from Ref. [13\*] is the repeatable avalanche breakdown. In Ref. [13\*], the breakdown is destructive breakdown, and the associated breakdown voltage is obtained in the static  $I$ - $V$  sweep. In comparison, the diode in this work undergoes repeatable avalanche breakdown, which is extremely essential for practical applications. The avalanche breakdown voltage in this work is evaluated under not only the static  $I$ - $V$  sweep test but also the dynamic test in the UIS circuit. As shown in Fig. R3 [Fig. 2(a) in our revised manuscript], the  $BV$  and the  $E_{\text{peak}}$  obtained in this work are stable and reliable under dynamic switching condition, which is consistent with commercial SiC diodes. The reason why we emphasize this “dynamic  $BV$ ” is that it has been recently reported to significantly differ from the “static  $BV$ ” in many devices. For example, in Ref. [17], the dynamic  $BV$  of a commercial GaN power device is found to be around 1.2 kV, which is only 40% of the static  $BV$  (nearly 2 kV), producing a remarkable voltage gap of 800 V. Therefore, for the fair comparison of our result to some other reported  $\text{Ga}_2\text{O}_3$  devices, the repeatable breakdown capability and the difference between static and dynamic  $BV$ s should be taken into account.

Fig. R2. Benchmark of  $E_{\text{peak}}$  versus  $BV$  for the large-sized, Ampere-class  $\beta\text{-Ga}_2\text{O}_3$ -based SBD, JBS diode, and HJD as well as the reports from Ref. [13\*]. Note that the device in Ref. [13\*] is small-area device.

Fig. R3.  $I$ - $V$  characteristics that combine the quasi-static  $I$ - $V$  curves (at low current levels) and the  $I_{\text{AVA}} \sim BV_{\text{AVA}}$  data obtained from the UIS tests, where both the HJD and the referenced commercial SiC diode show a smooth transition between the two sets of data.

**Comment (3):** Several  $\text{NiOx}/\text{Ga}_2\text{O}_3$  bipolar HJD's have been reported so far, whereas Fig. 5a only mentions a couple of data points (I believe the reason being Ampere-class). However, I think the overall contact area eventually determines the current levels, and hence I feel other reports also deserve a spot in Fig. 5a. [Kindly see the following papers: 10.1109/TPEL.2021.3123940, 10.1109/TED.2021.3091548].

**Reply:** Thanks for the kind reminder. While the critical difference between small-area and large-area devices has been explained in the last response, we agree with the reviewer that a comparison with all state-of-the-art reported devices (regardless of area) would be more comprehensive and make the conclusion more inclusive. The data for  $R_{\text{on,sp}}$  versus  $BV$  in the suggested publications [18], [19] have been included and summarized in Fig. R4, which has also been added to the supplementary Section S9 of the revised manuscript. In the same supplementary section, we also added a short discussion to illustrate the difference between small-area and large-area (ampere-class) devices,

which will help readers in the broad fields to understand their different implications for the Ga2O3 power device technology toward practical applications “...Note that ampere-level large-area power devices are required for industrial applications, so their performance usually represents the application prospects of a power device technology. For the emerging Ga2O3 power technology, there is still a large gap between the performance of large area devices and small-area devices, possibly due to the non-uniformity in fabrication process and wafer material properties. In addition, the breakdown voltage measured under the switching circuit tests instead of static I-V tests represents the true overvoltage margin of devices in practical power electronics applications. Note that, in this benchmark, the BV of devices in this work is the only one that has been validated in the circuit tests. The robust breakdown capability achieved in this work is attributed to the effective edge termination that incorporates the small-angle beveled junction termination extension (JTE) and high-k field plate...”.

Fig. R4.  $R_{on,sp}$  versus  $BV$  benchmark for Ga2O3 SBDs, JBS diodes and HJDs [2]-[15], [18], [19].

**Comment (4):** The authors claim hole injection into the Ga2O3 layer during forward bias (Fig. 4e). However, it is not clear how holes are able to overcome the large  $V_B$  barrier (authors claim a 3.2 eV  $V_B$ -offset between NiO and Ga2O3). Kindly explain. **Comment (5):** Kindly provide the barrier height and depletion width information in Fig. 4d. If the hole injection is tunneling assisted, kindly provide the tunneling probability of holes in overcoming the  $V_B$  barrier potential during forward bias.

**Reply:** We greatly appreciate the reviewers’ comments and questions. As both comments are related to the carrier transport under the device forward bias, we would like to make responses together. In the original manuscript, the Fig. 4(d) is a schematic illustration of band diagram at forward bias. To demonstrate the interfacial band bending more accurately, the quantitative band structure extracted from the simulation has been shown in Fig. 4(d) to replace the original band diagram schematic. The simulation results are detailed below, as well as the calculation of electron and hole tunneling probabilities.

Fig. R5. Simulated band structure near the NiO/Ga2O3 interface at a forward voltage of 6 V.

As the forward bias of the NiO/Ga2O3 PN junction increases, electron and hole accumulations are present at the Ga2O3 and NiO side of the heterojunction, respectively. Fig. R5 shows the simulated band structure near the heterojunction interface under a forward bias of 6 V. Note that the energy distance between the electron and hole quasi-Fermi levels are smaller than 6 eV near the junction due to the voltage drops in the drift region and substrate. The quantitative band structures, including the band bending and band offsets, allow the calculation of carrier tunneling probabilities across the interface in the conduction band and valance band, respectively (as illustrated by the arrows in Fig. R5). The zoom-in of the simulated band diagram, as shown in Fig. R6, suggests that the potential function near the interface can be approximated by an exponential function, as given by

$$U(x) = a \exp(bx) \quad (1)$$

Fig. R6. Schematic diagram of tunneling across an exponential-like potential barrier.

To calculate the tunneling probability, the wave function  $\Psi$  is determined by the Schrödinger equation:

$$\frac{d^2\Psi}{dx^2} + \frac{2m^*}{\hbar^2} [E - U(x)]\Psi = 0 \quad (2)$$

If the potential  $U(x)$  does not change abruptly, the Schrödinger equation can be approximated using the WKB (Wentzel-Kramers-Brillouin) method. In this case, the general form of the wave function is described as  $\exp(\int ik(x)dx)$ , where  $k = \sqrt{2m^*[E - U(x)]}/\hbar$ . Here  $E$  represents the carrier energy. As shown in Fig. R5, we set the zero energy for electrons as the conduction band minimum of NiO (i.e., the region for electrons to be tunneled to), and the zero energy for holes as the valence band minimum of Ga2O3. The electron energy and hole energy are denoted as  $E_e$  and  $E_h$ , respectively. The calculated tunneling probability can be written as [20],

$$T_t = \frac{|\Psi_B|^2}{|\Psi_A|^2} \approx \exp\left\{-2\int_0^{x_1} |k(x)| dx\right\} \approx \exp\left\{-2\int_0^{x_1} \sqrt{\frac{2m^*}{\hbar^2} [U(x) - E]} dx\right\} \quad (3)$$

Substituting Eq. (1) into Eq. (3), the tunneling probability is derived as,

$$T_t \approx \exp\left[-2\int_0^{x_1} \sqrt{2m^* [ae^{bx} - E]} / \hbar dx\right] \approx \exp\left[-\frac{4}{3b\hbar} \sqrt{2am^*} \left(e^{bx} - \frac{E}{a}\right)^{3/2} \Big|_0^{\frac{\ln(E/a)}{b}}\right] \quad (4)$$

where  $x_1$  is the distance when  $U(x_1) = E$ , i.e.,  $x_1 = \ln(E/a)/b$ .

Herein, the tunneling probabilities of electron (from Ga2O3 to NiO) and hole (from NiO to Ga2O3) has been calculated at different  $E_e$  and  $E_h$ , with the purpose of comparing the tunneling probabilities of electrons and holes. For this calculation, the exponential function is fitted against the simulated potential using Eq. (1), and the fitted parameters are listed in Table I, in which, the effective electron mass in NiO and the effective hole mass in Ga2O3 are also listed. Note that the hole effective mass in  $\beta$ -Ga2O3 still lacks a widely accepted value, as the reported values are scattered in a wide range from 1.50 to 267.38  $m_0$  [21-24]. Nevertheless, it is a consensus that  $\beta$ -Ga2O3 exhibits a very flat valence band, resulting in large hole effective mass ( $m_h$ ). Here, we choose a value of 5.30  $m_0$  according to [22] for the calculation of the hole tunneling probability.

**Fig. R7 presents the calculated tunneling probability results for  $E_e$  and  $E_h$  ranging from 0 to 1 eV. The results suggest that the electron tunneling probability is significantly higher than the hole tunneling probability throughout the entire energy ranges. This accords with the expectation from the lower conduction band offset as compared to the valence band offset (lower by ~1 eV).** It is worth noting that possible influence of interface traps is not taken into account in the calculation of carrier tunneling probabilities. In the presence of trap-assisted tunneling, the tunneling probabilities could be much higher than the calculated results. It is difficult to estimate the

probability of trap-assisted tunneling, as the nature, energetics and dynamics of the traps at the NiO/Ga2O3 interface as well as their roles in tunneling have not been fully understood in the literature. Whereas, we believe the above calculations are still useful to gain a comparative understanding of the electron and hole tunneling across the hetero-interface barriers.

Table I. Parameters used in the calculation of carrier tunneling probabilities

| $m_e$ in NiO    | $m_h$ Ga 2 O 3 | $a_{e\_NiO}$ (eV) | $b_{e\_NiO}$         | $a_{h\_GaO}$ (eV) | $b_{h\_GaO}$         |
|-----------------|--------------------------------------|-------------------|----------------------|-------------------|----------------------|
| 1.57 $m_0$ [25] | 5.30 $m_0$ [22]                      | 1.358             | $-1.151 \times 10^9$ | 1.957             | $-7.856 \times 10^8$ |

Fig. R7. Tunneling probabilities of the electron and hole versus  $E_e$  (or  $E_h$ ) at a forward voltage of 6 V.

Fig. R8. Tunneling probabilities of electrons and holes versus  $E_e$  (or  $E_h$ ) at a forward voltage of 6 V.

In the revised manuscript, we have updated the original Fig. 4(d) with the simulated band structure, as shown in Fig. R5. Additionally, the original Fig. 4(e) (Fig. R8 above) is updated accordingly to show the electron/hole distributions at the same 6 V forward bias condition. The simulation was calibrated with experiment through the current density. In the first step, we calibrated the interface recombination model by using the experimental current density at a subthreshold forward bias range (2 ~ 3 V), as interface recombination is the dominant mechanism in this

biasing regime. At a forward bias of 6 V, the experimental current is higher than the interface recombination current, as shown in the upwards  $I$ - $V$  curve in the original Fig. 3f, and the excessive current is ascribed to the diffusion of the tunneled electrons and holes. Secondly, we set the ratio of the electron and hole tunneling probabilities calculated in Fig. R7 while calibrating their absolute values (considering the trap-assisted tunneling) to make the total simulated current match the experimental values. As shown in the Fig. R8, the concentration of tunneled holes is much lower than that of tunneled electrons, which is consistent with the above tunneling probability calculations. In addition, the simulation further confirms the electron and hole accumulations at the hetero-junction interface.

**Comment (6):** *The authors claim "textbook-like avalanche" for Fig. 2a. However, the textbook model also talks about reverse saturation current increasing by 2x in every 10 degree rise in temperature. The net increase in the reverse current is less than 2x when temperature is increased from 25 C to 175 C. I am wondering what may be the cause of this increase, considering the bandgap may be high enough to prevent thermal generation. Kindly provide a justification for this in the manuscript so that the general audience is aware of the overall scenario.*

**Reply:** Thank you for the insightful comment. Indeed, the “textbook model of avalanche” in our original manuscript refers to avalanche waveforms instead of leakage current characteristics. The UIS waveform of HJD features avalanche characteristics typically described in the textbook. As  $I_{AVA}$  decreases from 30 A to zero, the voltage clamps at  $BV_{AVA}$ , and the energy stored in  $L_{UIS}$  is fully dissipated in the HJD during the 20- $\mu$ s avalanche process. However, the net increase in the reverse current is less than 2x when temperature is increased from 25 to 175 °C, which may result from the prevention of thermal generation due to its ultrawide bandgap property, as pointed out by the reviewer.

In the revised manuscript, to avoid the possible confusion, we have removed the phrase of “textbook-like avalanche”.

Here in the response, we expand some discussions on leakage current mechanisms.

In general, the  $I$ - $V$  characteristics of a p-n junction is expressed as [14], [26]:

$$I = -qAn_i \left( \frac{n_i}{N_D} \sqrt{\frac{D}{\tau}} + \frac{W}{2\tau} \right) \quad (4)$$

where  $n_i$  is the intrinsic carrier concentration,  $A$  is the area of the p-n diode,  $N_D$  is the n-type doping density,  $W$  is the width of the junction depletion region at applied voltage,  $D$  is the hole diffusion constant, and  $\tau$  is the effective minority carrier lifetime. The first term in the right side of Eq. (4) represents the diffusion current component contributed by minority carriers, while the second term arises from the thermal generation of carriers in the depletion region. The calculated reverse leakage current of NiO/Ga2O3 HJD related to thermal generation is shown in Fig. R9. The reverse current density at a reverse bias of 1000 V and a temperature of 175 °C is only 10-22 A/cm2, which has

negligible contribution to the total leakage current. It is attributed to the large bandgap of Ga2O3 with a rather low intrinsic carrier concentration, as pointed out by the reviewer.

Fig. R9. Reverse leakage current of NiO/Ga2O3 HJD based on thermal generation model.

As reported in our previous work, trap-assisted tunneling (TAT) is a dominant contributor to the leakage current. The expression of trap assisted tunneling current is given by [27], [28]

$$I = A_{\text{TAT}} \exp\left(\frac{-8\pi\sqrt{2qm^*}}{3hE} \phi_t^{3/2}\right) \quad (5)$$

where  $m^*$  is the electron effective mass of Ga2O3 ( $m^* = 0.34 m_0$ ) [29],  $A_{\text{TAT}}$  is a generic constant,  $h$  is Planck's constant, and  $\Phi_t$  is the energy of the electron traps. Fig. R10(a) shows the  $\ln(J)$  of experimental data plotted as a function of  $1/E$ . A linear relationship is observed, implying that the leakage current is mainly resulted from trap-assisted tunneling. The corresponding schematic diagram of electron tunneling from the valence band of NiO to the conduction band of Ga2O3 is shown in Fig. R10(b). The energy level of traps is derived to be 0.11 eV below conduction band edge. It is consistent with the energy level of incompletely ionized Si shallow donors in Ga2O3 (0.10 ~ 0.12 eV) [30], [31].

Fig. R10. (a) Plot of the  $\ln(J)$  vs.  $1/E$  with temperature increasing from 25 to 175 °C. (b) Illustration of carrier transport path in the framework of TAT mechanism under high electric field.

## **II. Reply to the 2nd reviewer's comments**

*The present work is related to very hot topic related to the new generation ultra-wide band gap material - Ga2O3 potential application into power electronics. Authors present the study of avalanche and surge robustness study under extremely high electric field and current density; this knowledge is very important for development power devices based on Gallium Oxide. The main achievement is:*

*-Give insights into the fundamentals of hole transport dynamics in Ga2O3*

*- Estimation experimentally impact ionization coefficients*

*-The results open a new pathway for making robust Ga2O3/NiO heterojunction-based power devices*

**Comment (1):** *Author should not state as first original demonstration/measurements of hole carrier diffusion length and life time determination. Such values were reported in Appl. Phys. Lett. 112, 082104 (2018); doi: 10.1063/1.5011971; 10.1149/2.0101702jss; APL Materials 10, 031106 (2022); <https://doi.org/10.1063/5.0086449>*

**Reply:** Thanks for the kind correction. As suggested, the statement on “first original demonstration/measurement of the hole carrier diffusion length and lifetime determination” is removed in the revised manuscript, as: “...Accordingly, the minority carrier diffusion length (L) for electrons in p-NiO and holes in n-Ga2O3 are determined to be 413 and 127 nm, respectively (Fig. 4c). Consequently, the minority carrier lifetime (τ) for electrons in p-NiO and holes in Ga2O3 are extracted to be 124.0 and 6.2 ns, respectively...”. Please see the changes in the 2nd paragraph of “Microscopic EBIC Characterization” section in the revised manuscript.

**Comment (2):** *There are lack of information/precisions in the part related to device fabrication Process S1: What is the origin (how are they fabricated?) of conductive Sn-doped (001) β-Ga2O3 substrates? What is the Sn doping level and electrical resistivity? What was the growth temperature of Si: Ga2O3 epilayers? What is the Si doping level (exact concentration) in Ga2O3 epi-wafers? How  $n=1.7 \times 10^{16} \text{ cm}^{-3}$  electron concentration for epilayers was measured? And what is the mobility value?*

**Reply:** Thanks for the comment. As suggested, we have added more detailed descriptions in the part related to device fabrication process S1 in the revised manuscript and supplementary materials. The β-Ga2O3 (001) epi-wafer used in this work was from the commercial vendor in Japan, the Novel Crystal Technology (NCT). The datasheet provided by the NCT shows that the substrate is the conductive Sn-doped (001) β-Ga2O3 grown by edge-defined film-fed growth (EFG) technique with an electron concentration of about  $6 \times 10^{18} \text{ cm}^{-3}$  and a resistivity of  $5 \times 10^{-2} \Omega \cdot \text{cm}$ . A 10-μm-thick lightly-doped homoepitaxial layer was grown by halide vapor phase epitaxy (HVPE) technique by NCT [32]. For the doped film with the carrier density of  $1 \times 10^{16} \text{ cm}^{-3}$ , the activation energy and the mobility at room

temperature were 45.6 meV and 145 cm2/V·s, respectively, as determined by the temperature-dependent Hall measurement [32]. The depth profile of the doping concentration has been confirmed by performing capacitance-voltage (*C-V*) experiments. As shown in Fig. R11, the derived net doping concentration ( $N_D - N_A$ ) is averaged to be about  $1.7 \times 10^{16}$  cm-3.

Fig. R11. The depth profile of ( $N_D - N_A$ ) in the n-Ga2O3 drift layer extracted from *C-V* curve.

**Comment (3):** Regarding Annealing a) If the goal of annealing as it is stated was to decrease donor concentrations in Si:Ga2O3 to have resistive layer why epilayers were doped during the growth? It is not written what is the optimal electron concentration targeted by authors for this device structure. b) Annealing at 500 °C for 5 hours under the O2 ambient: can occur the inter-diffusion of Sn from substrate and Si from epi-wafer? How changes (if this is a case) interface profile after annealing? It is not addressed this issue by authors. What was the electron concentration value after annealing? Can Ga2O3:Si epilayer be still considered as an n type or it becomes semi-insulating?

**Reply:** Thanks for the comment. In recent years, it has been reported that near-surface defects in Ga2O3 have profound impacts on the Schottky barrier height and electrical properties of the Ga2O3 device [26], [33-35]. To passivate such surface defects, the low-temperature oxygen annealing pretreatment has been proved to be an effective process solution [33]. **Thus, the optimized annealing at 500 °C in O2 for 5 hours in this work is to passivate the possible surface defects.**

Furthermore, according to literature reports [36], [37], [38], [39], we don't expect the inter-diffusion of Sn from substrate and Si from epi-wafer under such a low annealing temperature. The diffusivity is often used to indicate the inter-diffusion ability of dopants, which depends on doping concentration and annealing conditions. Based on experimental and simulation results, Ribhu Sharma *et al.* have reported that the diffusivity of Sn and Si in O2 ambient at 1150 °C is in the range of  $2.7 \sim 9.5 \times 10^{-13}$  cm2/s, significantly lower than that of Ge dopant

**Editorial Note:** Figure below from Murakami, H. et al. Homoepitaxial growth of  $\beta$ -Ga2O3 layers by halide vapor phase epitaxy. *Appl. Phys. Exp.* **8**, 015503 (2015), doi: 10.7567/APEX.8.015503. © The Japan Society of Applied Physics. Reproduced by permission of IOP Publishing Ltd. All rights reserved.

( $\sim 1.1 \times 10^{-11}$  cm2/s) [36]. Ymir K. Frodason *et al.* have also observed a sharp drop in the concentration at the interface between the epitaxial layer and the Sn-doped substrate by treating the samples at different temperatures. Evident diffusion of Sn dopant occurred only when the temperature reached 1150~1250 °C [38]. It was also reported that, during the HVPE growth at a high temperature of 1000 °C, only relatively limited inter-diffusion of Sn and Si dopants occurs, as confirmed by Secondary ion mass spectrometry (SIMS) characterizations in Fig. R12 [39].

In summary, the low-temperature annealing at 500 °C for 5 hours in the O2 ambient is favorable to passivate near-surface defects in the HVPE-grown epilayer, while the inter-diffusion of Sn and Si dopants are negligible. In the revised manuscript, the effect of low temperature annealing has been discussed in the “Methods” section as “...This annealing process has been reported to be able to effectively passivate near-surface defects in Ga2O3 epilayer [33]. Note that the inter-diffusion of Sn from substrate and Si from epi-wafer is expected to be negligible at this annealing temperature [38], [39]...”.

Fig. R12. SIMS depth profiles of an HVPE layer grown at 1000 °C reported in [39]. Arrows represent the background concentrations of the elements.

**Comment (4):** It is not mentioned what is I-V characteristic of Ti/Au (20/80 nm) metal contacts deposited back side on substrate? It is not discussed, how were measured hole concentrations in NiO layers? What is hole mobilities? Authors do not comment obtained values of impact ionization coefficients for holes and electrons. Why I.I. coefficient is higher for holes? How it correlates with low hole mobilities and higher (than for electrons) effective mass? Will

be good to compare with impact ionization coefficients for SiC measured experimentally.

**Reply:** We greatly appreciate the reviewers' comments and questions. Firstly, supplementary experiments on the  $I$ - $V$  characteristics of Ti/Au (20/80 nm) metal contacts deposited on the backside of the substrate have been done. Fig. R13(a) shows the circular transmission line method (CTLM) measurement results. The transport properties in the metal/semiconductor contacts including the transfer length ( $L_T$ ), contact resistance ( $R_c$ ), the specific contact resistance ( $\rho_c$ ), and the sheet resistance ( $R_{sh}$ ) are determined. The  $I$ - $V$  curves of the metal/semiconductor contacts show a linear increase over the entire  $L_T$  range from 5 to 35  $\mu\text{m}$ , revealing its Ohmic conduction nature of the Ti/Au (20/80 nm) on the backside of the Sn-doped  $\text{Ga}_2\text{O}_3$  substrate. Fig. R13 (b) shows the extracted  $R_c$  as a function of  $L_T$ , and the  $\rho_c$  and  $R_{sh}$  are estimated to be  $0.156 \text{ m}\Omega\cdot\text{cm}^2$  and  $15.5 \text{ }\Omega/\text{sq}$ , respectively, in terms of contact perimeter and area of  $314 \text{ }\mu\text{m}$  and  $7.875 \times 10^{-5} \text{ cm}^2$ .

Secondly, NiO films with controlled hole concentrations are simultaneously deposited on the semi-insulating Fe-doped  $\text{Ga}_2\text{O}_3$  substrates by RF magnetron sputtering technique at room temperature for Hall measurements. By performing hall measurements on these controlled samples at room temperature, hole mobilities corresponding to p-NiO ( $5.8 \times 10^{17} \text{ cm}^{-3}$ ) and p+-NiO ( $2.9 \times 10^{19} \text{ cm}^{-3}$ ) layers are  $0.87 \text{ cm}^2/\text{V}\cdot\text{s}$  and  $0.34 \text{ cm}^2/\text{V}\cdot\text{s}$ , respectively.

Fig. R13. (a) The  $I$ - $V$  curves of the Ti/Au metal contacts deposited on the backside of the substrate with different  $L_T$ s. The inset is top-view microscopy images of CTLM structures. (b) The  $R_c$  as a function of the  $L_T$ .

On the other hand, the electron impact ionization coefficient ( $\alpha_n$ ) and hole impact ionization coefficient ( $\alpha_p$ ) used in this work were obtained by fitting the experimental avalanche data (in the Supplementary section S4). In the fitting, we used the value of  $\alpha_n$  predicted in Ref. [40]. Note that the impact ionization coefficients ( $\alpha_n$  and  $\alpha_p$ ) are not directly dependent on the carrier mobility and effective mass, but strongly dependent on the electric field ( $\mathcal{E}$ ). To

illustrate the key determining factors, we refer to the Thornber model [41], which is a widely-used semi-empirical model for impact ionization, as given by,

$$\alpha \approx \frac{q\xi}{E_i} \exp \left( - \frac{\frac{3}{2} E_G}{q\xi\lambda + \frac{q^2 \xi^2 \lambda^2}{E_r} + E_{kb} T} \right) \quad (6)$$

where  $E_G$ ,  $\xi$ , and  $E_r$  are the material bandgap, electric field, and optical phonon energy, respectively.  $\lambda$  is the mean free path for electrons or holes, which is approximately the average distance that a charge carrier travels before being scattered by phonons, impurities, or impact-ionization events.  $E_i$  represents the field to achieve an ionizing collision, and  $E_{kb}$  represents the energy due to thermal effects. For the calculation of  $\alpha_n$  and  $\alpha_p$ , major parameters of various WBG and UWBG materials from Ref. [41] are listed in Table II. Holes generally show a smaller  $\lambda$  due to a high effective mass but possess a higher impact ionization coefficient owing to other dominant parameters ( $E_G$ ,  $\xi$ ,  $E_r$ ,  $\lambda$ ,  $E_i$  and  $E_{kb}$ ). Similarly, it was reported by many experimental results that  $\alpha_p$  is higher than  $\alpha_n$  for GaN and SiC materials, as summarized in Fig. R14 [42]. This showcases that impact ionization coefficient is not dominated by the effective mass or carrier mobility.

Table II. Parameters for the Thornber model of impact ionization coefficients at 300 K [41]

| Parameters                      |          | Si    | 4H-SiC | GaN    | $\beta$ -Ga 2 O 3 | Al 0.6 Ga 0.4 N | Diamond | AlN  |
|---------------------------------|----------|-------|--------|--------|-----------------------------------------|---------------------------------------|---------|------|
| $\lambda$ (10 -10 m) | Electron | 67.5  | 10.0   | 3.46   | 26.1                                    | 16.7                                  | 48.8    | 22.6 |
|                                 | Hole     | 25.1  | 4.80   | 2.83   | —                                       | —                                     | 57.1    | —    |
| $E_i$ (eV)                      | Electron | 2.51  | 7.50   | 3.43   | 7.35                                    | 7.12                                  | 165     | 12.0 |
|                                 | Hole     | 3.06  | 6.62   | 1.75   | —                                       | —                                     | 4.94    | —    |
| $E_r$ (eV)                      | Electron | 0.106 | 0.0925 | 0.0175 | —                                       | 0.879                                 | 2.17    | 1.42 |
|                                 | Hole     | 0.021 | 0.0090 | 0.0070 | —                                       | —                                     | 1.94    | —    |
| $E_{kBT}$ (meV)                 | Electron | 49.4  | 14.4   | 279    | 18.8                                    | 26.2                                  | 5.59    | 24.7 |
|                                 | Hole     | 12.0  | 102    | 196    | —                                       | —                                     | 5.88    | —    |

**Editorial Note:** Figure below © 2023 IEEE. Reprinted, with permission, from J. A. Cooper and D. T. Morissette, "Performance Limits of Vertical Unipolar Power Devices in GaN and 4H-SiC," in *IEEE Electron Device Letters*, vol. 41, no. 6, pp. 892-895, June 2020, doi: 10.1109/LED.2020.2987282.

Fig. R14. Impact ionization coefficients for holes ( $\alpha_p$ ) and electrons ( $\alpha_n$ ) in (a) SiC and (b) GaN at 27 °C from multiple reports [42].

As suggested, supplementary statements on the Ohmic contact properties of backside metals, hole concentration and mobility, and impact ionization coefficients have been added as in the revised manuscript “...By performing capacitance-voltage (C-V) experiments, the net doping concentration is averaged to be  $1.7 \times 10^{16} \text{ cm}^{-3}$ . The mobility value of epilayer was reported to be  $145 \text{ cm}^2/\text{V}\cdot\text{s}$  with a carrier concentration of  $\sim 10^{16} \text{ cm}^{-3}$ , as measured by Hall test at room temperature [32]. The main fabrication steps of the NiO/Ga2O3 HJDs include (a) substrate thinning and wafer cleaning, (b) cathode metal deposition...” in the Supplementary Section S1, and “...Based on the hall measurements of the controlled NiO samples on semi-insulating substrates, hole mobilities corresponding to p-NiO ( $5.8 \times 10^{17} \text{ cm}^{-3}$ ) and p+-NiO ( $2.9 \times 10^{19} \text{ cm}^{-3}$ ) layers were determined to be  $0.87 \text{ cm}^2/\text{V}\cdot\text{s}$  and  $0.34 \text{ cm}^2/\text{V}\cdot\text{s}$ , respectively...” in the Methods section, and “...Fig. S9(a) suggests that the I. I. coefficient of hole is higher than that of electron in Ga2O3 drift region, which is consistent with the cases in SiC and GaN materials [42]. Note that the I. I. coefficient is influenced by multiple material properties including the material bandgap, electric field, optical phonon energy, mean free path, electron mobility and electric field, as suggested by the widely-used Thornber model [41]...” in the Supplementary Section S4.

**Comment (5):** I-Page.4 Ref.23 will be good to give corresponding valued for SiC for comparison. Page;5 regarding simulation where input is  $\mu_p$  values. Authors could take values for hole mobilities which are experimentally measured in undoped Ga2O3 thin films from for example: *Journal of Materials Chemistry C*, 2019, 7, 10231 <https://doi.org/10.1039/C9TC02910A>; *Materials Today Physics*, 3, 118; (2017) <https://doi.org/10.1016/j.mtphys.2017.10.002> ( $\mu=8\text{-}10 \text{ cm}^2/\text{V}\cdot\text{s}$ ) value could better explain perhaps avalanche in n-Ga2O3.

In the model for avalanche simulation, Table S2: Will be more interesting and realistic to use data from experiment,

*i.e. measured values of mobilities.*

*Supplementary section S1: in all figures, substrate and epilayer doping will be good to indicate: i.e. Si:Ga2O3 and Sn:Ga2O3, indeed both of them are n-type.*

**Reply:** Thanks for these constructive suggestions. As suggested, the corresponding leakage value (~10  $\mu\text{A}$ ) for the commercial SiC SBD under a reverse bias of 1200 V have been added in the revised manuscript for comparison. We agree with the reviewer that the experimental data of hole mobilities are more suitable for use in simulations to discuss the avalanche mechanism. The hole mobility in the suggested Ref. [43], [44] are 8-10  $\text{cm}^2/\text{V}\cdot\text{s}$ , while that in Ref. [45] is ~1.2  $\text{cm}^2/\text{V}\cdot\text{s}$ . Considering the relative large spread of hole mobility in the experimental report, we choose a low hole mobility of 1  $\text{cm}^2/\text{V}\cdot\text{s}$  in this work as the “worst scenario” for avalanche. Higher hole mobility would allow more efficient hole removal, making it easier to achieve high avalanche current. In the revised manuscript, we added additional explanations to clarify this point as “...Note that higher  $\mu_p$  values, e.g., ~1.2  $\text{cm}^2/\text{V}\cdot\text{s}$  [45] and 8~10  $\text{cm}^2/\text{V}\cdot\text{s}$  [43], [44], have been reported experimentally. Here we use two lower  $\mu_p$  values in the simulation mainly to consider the worst scenario of avalanche, as a high  $\mu_p$  can allow for a more efficient hole removal and thus supports high avalanche current...” Please see the changes in the “Avalanche robustness” section in the revised manuscript.

On the other hand, key parameters used for the simulations in Table S2 are replaced by the reported experimental values, except for the minority carrier mobilities in n-Ga2O3 and p-NiO, both of which are difficult to obtain experimentally. Meanwhile, the doping levels of n-type Sn-doped Ga2O3 substrate and Si lightly-doped Ga2O3 epilayer have been added to Supplementary section S1.

**Comment (6):** *It is not very clear the connection of the present work with following publications:*

*\* $\beta$ -Ga2O3 hetero-junction barrier Schottky diode with reverse leakage current modulation and  $BV^2/R_{on,sp}$  value of 0.93  $\text{GW}/\text{cm}^2$  Appl. Phys. Lett. 118, 122102 (2021); doi: 10.1063/5.0044130*

*\*\* Zhang, J. et al. Ultra-wide bandgap semiconductor Ga2O3 power diodes. Nat. Commun. 13, 3900(2022). It should be corrected cited those articles and underlined the similarity and the difference with the results in present manuscript. - Hsiao-Hsuan Wan, J. Vac. Sci. Technol. A 41, 032701 (2023) is very important to cite.*

*In Page 2: Authors state idea that p-type doping has been unsuccessful. That is wrong, since there are reports demonstrating native hole conductivity and Zn and N doping related hole conductivity as well. [Journal of Vacuum Science & Technology A 40, 043401, 2022. <https://doi.org/10.1116/6.0001766>; Journal of Materials Chemistry C, 2019, 7, 10231 <https://doi.org/10.1039/C9TC02910A> <https://doi.org/10.1016/j.mtphys.2021.100356>].*

*Though and evidently, reported hole concentrations are not enough for device functioning.*

It is interesting to compare carrier diffusion length for Ga2O3 reported in Applied Materials letters, APL 10, 031106 (2022); <https://doi.org/10.1063/5.0086449>

-Minority (holes) diffusion length and lifetimes for Ga2O3 have been already reported by authors in: Appl. Phys. Lett. 112, 082104 (2018); doi: 10.1063/1.5011971

Authors should cite this publication and compare their results.

It was reported experimentally estimated breakdown electrical field for Ga2O3 in the <https://doi.org/10.1016/j.mtphys.2020.100263>, will be interesting if authors will make comment regarding.

**Reply:** Thanks for the comments. For the suggested references [16], [46], [47], [48], high breakdown voltages or large breakdown electric fields have been reported in small-area power diodes with low forward current at mA level. In contrast, power devices for practical power applications usually require a minimal current rating of at least several amperes (more common is a current rating of tens of amperes even up to hundreds of amperes), so the ampere-class device performance represents the true state-of-the-art of a power device technology. For the emerging Ga2O3 device technology, there is a large gap between the performance of large-area and small-area devices, possibly due to the nonuniformities in fabrication process and material properties. Fig. R15 below depicts the  $E_{\text{peak}}$  versus  $BV$  for reported large-area Ga2O3 rectifiers (ampere-class), including SBDs, HJDs and junction barrier Schottky diode (JBSs) [2-16]. Overall, the  $E_{\text{peak}}$  of most reported Ga2O3 diodes is below 3 MV/cm. The  $E_{\text{max}}$  of about 4.6 MV/cm (see Fig. 1e) achieved in this work highlights one of the highest  $E_{\text{peak}}$  reported in ampere-class Ga2O3 devices.

Fig. R15. Benchmark of  $E_{\text{peak}}$  versus  $BV$  for reported large-area Ga2O3 SBDs, HJDs and junction barrier Schottky diode (JBSs).

More important than device performance, the achievement in this work differs from other reported state-of-the-art Ga2O3 diodes in the following aspects.

- 1) Avalanche and surge robustness are the essential prerequisite of any power semiconductor device to survive the common overvoltage and overcurrent stresses in power electronics applications. Compared to traditional homogenous p-n junctions, the robust heterojunction constructed in this research proves the fundamental viability to achieve robustness in heterogeneous junctions and show a feasible way to achieve robustness in UWBG power devices (which usually lacks the intrinsic bipolar doping). This new finding is expected to be greatly impactful in the broader range of applied science and engineering fields.
- 2) Carrier transport under extreme conditions such as high electric field, high current density, and high temperature, as well as non-equilibrium dynamic condition, is involved in avalanche and surge operations. In this work, we use the dynamic avalanche characteristics to show, for the first time, the hole's exemption from the controversial self-trapping effect in Ga2O3 under high field and extract experimentally one of the first full sets of electron and hole impact ionization coefficients in Ga2O3.
- 3) We show the interesting carrier dynamics (e.g., conductivity modulation in p-NiO instead of n-Ga2O3) can result in a new UWBG heterogeneous device that is incredibly robust at the same time switches at a significantly faster speed as compared to some conventional homo-junctions (e.g., in Si). Such breakthrough performance regarding the trade-off between robustness and switching speed has never been reported in Ga2O3 devices before.

Furthermore, regarding the reports of p-type Ga2O3 in literature [44], [49], [50], the conduction mechanisms of holes are mainly through hopping conduction rather than band transport. In this circumstance, the claimed p-type conduction is mainly evaluated by material characterization, and the function was not confirmed in the device aspect. As far as we know, there still lacks reports of high-performance Ga2O3 devices relying on the intrinsic p-type Ga2O3. Therefore, although p-type Ga2O3 have been reported by some groups, the reliability and stability are still a big challenge, which is widely accepted by the Ga2O3 research community. Not to misguide readers, we change the relevant statement as “...due to the flat valence band and strong self-trapping of holes, the reliable p-type doping in Ga2O3 is very challenging, although p-type Ga2O3 have been reported by some groups [44], [49], [50]...”.

In addition, the reviewer raises the concern on the differences of the minority carrier diffusion lengths in this work and other reported values. Indeed, in the suggested Ref. [51], [52], the minority carrier diffusion lengths for electrons in p-Ga2O3 and holes in n-Ga2O3 at ~300 K have been experimentally determined to be 1040 and 325 nm, respectively, which are different with the experimental values of 413 and 127 nm by using EBIC technique in this work. We would like to state that, although both results are determined by the same method of EBIC, such difference

results from the variation in epitaxial quality, material synthesis methods, and material conductivities. In the revised manuscript, a comment on the minority carrier diffusion length has been added as “...Accordingly, the minority carrier diffusion length ( $L$ ) for electrons in  $p$ -NiO and holes in  $n$ -Ga2O3 are determined to be 413 and 127 nm, respectively (Fig. 4c). The difference to the reported values in [51], [52] could be resulted from the variation in epitaxial quality, synthesis methods, and material conductivities...”.

## References

- [1] Krtschil, A., Dadgar, A. & Krost, A. Decoration effects as origin of dislocation-related charges in gallium nitride layers investigated by scanning surface potential microscopy. *Appl. Phys. Lett.* **82**, 2263-2265 (2003).
- [2] Zhou, F. *et al.* 1.95-kV Beveled-mesa NiO/Ga2O3 Heterojunction Diode with 98.5% Conversion Efficiency and Over Million-Times Overvoltage Ruggedness. *IEEE Trans. Power Electron.* **37**, 1223-1227 (2021).
- [3] Gong, H. *et al.* 1.37 kV/12 A NiO/ $\beta$ -Ga2O3 heterojunction diode with nanosecond reverse recovery and rugged surge-current capability. *IEEE Trans. Power Electron.* **36**, 12213-12217 (2021).
- [4] Gong, H. *et al.* 70- $\mu$ m-Body Ga2O3 Schottky Barrier Diode with 1.48 K/W Thermal Resistance, 59 A Surge Current and 98.9% Conversion Efficiency. *IEEE Electron Device Lett.* **43**, 773-776 (2022).
- [5] Ji, M. *et al.* Demonstration of large-size vertical Ga2O3 Schottky barrier diodes. *IEEE Trans. Power Electron.* **36**, 41-44 (2021).
- [6] Lv, Y. *et al.* Demonstration of  $\beta$ -Ga2O3 Junction Barrier Schottky Diodes with a Baliga's Figure of Merit of 0.85 GW/cm2 or a 5A/700 V Handling Capabilities. *IEEE Trans. Power Electron.* **36**, 6179-6182 (2020).
- [7] Otsuka, F. *et al.* Large-size (1.7  $\times$  1.7 mm2)  $\beta$ -Ga2O3 field-plated trench MOS-type Schottky barrier diodes with 1.2 kV breakdown voltage and 109 high on/off current ratio. *Appl. Phys. Exp.* **15**, 016501 (2021).
- [8] Sharma, R. *et al.* Effect of probe geometry during measurement of >100 A Ga2O3 vertical rectifiers. *J. Vac. Sci. Technol., A* **39**, 013406 (2021).
- [9] Wei, J. *et al.* Experimental Study on Electrical Characteristics of Large-Size Vertical  $\beta$ -Ga2O3 Junction Barrier Schottky Diodes. In *2022 IEEE 34th International Symposium on Power Semiconductor Devices and ICs*. 97-100 (IEEE, 2022); doi: 10.1109/ispsd49238.2022.9813623.
- [10] Xiao, M. *et al.* Packaged Ga2O3 Schottky Rectifiers with Over 60 A Surge Current Capability. *IEEE Trans. Power Electron.* **36**, 8565-8569 (2021).
- [11] Yang, J. *et al.* Reverse Breakdown in Large Area, Field-Plated, Vertical  $\beta$ -Ga2O3 Rectifiers. *ECS J. Solid State Sci. Technol.* **8**, Q3159-Q3164 (2019).
- [12] Yang, J., Ren, F., Tadjer, M., Pearton, S. J. & Kuramata, A. Ga2O3 Schottky rectifiers with 1 ampere forward current, 650 V reverse breakdown and 26.5 MW.cm-2 figure-of-merit. *AIP Adv.* **8**, 055026 (2018).
- [13] Yang, J. *et al.* Vertical geometry 33.2 A, 4.8 MW cm2 Ga2O3 field-plated Schottky rectifier arrays. *Appl. Phys. Lett.* **114**, 232106 (2019).
- [14] Zhang, Y., Udrea, F. & Wang, H. Multidimensional device architectures for efficient power electronics. *Nat. Electron.* **5**, 723-734 (2022).
- [15] Zhou, F. *et al.* Over 1.8 GW/cm2 beveled-mesa NiO/ $\beta$ -Ga2O3 heterojunction diode with 800 V/10 A nanosecond switching capability. *Appl. Phys. Lett.* **119**, 262103 (2021).
- [16] Zhang, J. *et al.* Ultra-wide bandgap semiconductor Ga2O3 power diodes. *Nat. Commun.* **13**, 3900 (2022).
- [17] Song, Q. *et al.* Robustness of Cascode GaN HEMTs in Unclamped Inductive Switching. *IEEE Trans. Power Electron.* **37**, 4148-4160 (2022).

- [18] Wang, Y. g. *et al.* 2.41 kV Vertical p-NiO/n-Ga2O3 Heterojunction Diodes with a Record Baligas Figure-of-Merit of 5.18 GW/cm2. *IEEE Trans. Power Electron.* **37**, 3743-3746 (2022).
- [19] Luo, H. *et al.* Fabrication and Characterization of High-Voltage NiO/ $\beta$ -Ga2O3 Heterojunction Power Diodes. *IEEE Trans. Electron Devices* **68**, 3991-3996 (2021).
- [20] Sze, S. M., Li, Y. & Ng, K. K. *Physics of semiconductor devices.* (John wiley & sons, 2021).
- [21] Su, J. *et al.* Unusual electronic and optical properties of two-dimensional Ga2O3 predicted by density functional theory. *The Journal of Physical Chemistry C* **122**, 24592-24599 (2018).
- [22] Yuan, H. *et al.* Tuning the intrinsic electric field of Janus-TMDs to realize high-performance  $\beta$ -Ga2O3 device based on  $\beta$ -Ga2O3/Janus-TMD heterostructures. *Materials Today Physics* **21**, 100549 (2021).
- [23] Mock, A. *et al.* Band-to-band transitions, selection rules, effective mass, and excitonic contributions in monoclinic  $\beta$ -Ga2O3. *Phys. Rev. B* **96**, 245205 (2017).
- [24] Li, L., Liao, F. & Hu, X. The possibility of N-P codoping to realize P type  $\beta$ -Ga2O3. *Superlattices Microstruct.* **141**, 106502 (2020).
- [25] Fo, Y., Wang, M., Ma, Y., Dong, H. & Zhou, X. Origin of highly efficient photocatalyst NiO/SrTiO3 for overall water splitting: Insights from density functional theory calculations. *J. Solid State Chem.* **292**, 121683 (2020).
- [26] Hong, Y.-H. *et al.* The optimized interface characteristics of  $\beta$ -Ga2O3 Schottky barrier diode with low temperature annealing. *Appl. Phys. Lett.* **119**, 132103 (2021).
- [27] Houg, M. P., Wang, Y. H. & Chang, W. J. Current transport mechanism in trapped oxides: A generalized trap-assisted tunneling model. *Journal of Applied Physics* **86**, 1488-1491 (1999).
- [28] Guo, X. *et al.* Reverse leakage and breakdown mechanisms of vertical GaN-on-Si Schottky barrier diodes with and without implanted termination. *Appl. Phys. Lett.* **118**, 243501 (2021).
- [29] Sasaki, K., Higashiwaki, M., Kuramata, A., Masui, T. & Yamakoshi, S. Ga2O3 Schottky Barrier Diodes Fabricated by Using Single-Crystal  $\beta$ -Ga2O3 (010) Substrates. *IEEE Electron Device Lett.* **34**, 493-495 (2013).
- [30] Huang, S.-S. *et al.*  $\beta$ -Ga2O3 defect study by steady-state capacitance spectroscopy. *Japanese Journal of Appl. Phys.* **57**, 091101 (2018).
- [31] Ghadi, H. *et al.* Influence of growth temperature on defect states throughout the bandgap of MOCVD-grown  $\beta$ -Ga2O3. *Appl. Phys. Lett.* **117**, 172106 (2020).
- [32] Goto, K. *et al.* Halide vapor phase epitaxy of Si doped  $\beta$ -Ga2O3 and its electrical properties. *Thin Solid Films* **666**, 182-184 (2018).
- [33] Hu, H. *et al.* The role of surface pretreatment by low temperature O2 gas annealing for  $\beta$ -Ga2O3 Schottky barrier diodes. *Appl. Phys. Lett.* **120**, 073501 (2022).
- [34] Lingaparathi, R. *et al.* Surface states on (001) oriented  $\beta$ -Ga2O3 epilayers, their origin, and their effect on the electrical properties of Schottky barrier diodes. *Appl. Phys. Lett.* **116**, 092101 (2020).
- [35] Lingaparathi, R. *et al.* Effects of Oxygen Annealing of  $\beta$ -Ga2O3 Epilayers on the Properties of Vertical Schottky Barrier Diodes. *ECS J. Solid State Sci. Technol.* **9**, 024004 (2020).
- [36] Sharma, R., Law, M. E., Ren, F., Polyakov, A. Y. & Pearton, S. J. Diffusion of dopants and impurities in  $\beta$ -Ga2O3. *J. Vac. Sci. Technol., A* **39**, 060801 (2021).
- [37] Wong, M. H. *et al.* Acceptor doping of  $\beta$ -Ga2O3 by Mg and N ion implantations. *Appl. Phys. Lett.* **113**, 102103 (2018).
- [38] Frodason, Y. K. *et al.* Diffusion of Sn donors in  $\beta$ -Ga2O3. *APL Mater.* **11**, 041121 (2023).
- [39] Murakami, H. *et al.* Homoepitaxial growth of  $\beta$ -Ga2O3 layers by halide vapor phase epitaxy. *Appl. Phys. Exp.* **8**, 015503 (2015).
- [40] Ghosh, K. & Singiseti, U. Impact ionization in  $\beta$ -Ga2O3. *J. Appl. Phys.* **124**, 085707 (2018).
- [41] Nouketcha, F. L. *et al.* Investigation of wide-and ultrawide-bandgap semiconductors from impact-ionization coefficients. *IEEE Transactions on Electron Devices* **67**, 3999-4005 (2020).

- [42] Cooper, J. A. & Morisette, D. T. Performance limits of vertical unipolar power devices in GaN and 4H-SiC. *IEEE Electron Device Letters* **41**, 892-895 (2020).
- [43] Chikoidze, E. *et al.* P-type  $\beta$ -gallium oxide: A new perspective for power and optoelectronic devices. *Mater. Today Phys.* **3**, 118-126 (2017).
- [44] Chikoidze, E. *et al.* Enhancing the intrinsic p-type conductivity of the ultra-wide bandgap Ga2O3 semiconductor. *J. Mater. Chem. C* **7**, 10231-10239 (2019).
- [45] Ponc e, S. & Giustino, F. Structural, electronic, elastic, power, and transport properties of  $\beta$ -Ga2O3 from first principles. *Phys. Rev. Res.* **2**, 033102 (2020).
- [46] Yan, Q. *et al.*  $\beta$ -Ga2O3 hetero-junction barrier Schottky diode with reverse leakage current modulation and BV2/Ron,sp value of 0.93 GW/cm2. *Appl. Phys. Lett.* **118**, 122102 (2021).
- [47] Wan, H.-H. *et al.* NiO/ $\beta$ -(AlxGa1-x)2O3/Ga2O3 heterojunction lateral rectifiers with reverse breakdown voltage >7 kV. *J. Vac. Sci. Technol., A* **41**, 032701 (2023).
- [48] Chikoidze, E. *et al.* Ultra-high critical electric field of 13.2 MV/cm for Zn-doped p-type  $\beta$ -Ga2O3. *Mater. Today Phys.* **15**, 100263 (2020).
- [49] Wu, Z. Y. *et al.* Energy-driven multi-step structural phase transition mechanism to achieve high-quality p-type nitrogen-doped  $\beta$ -Ga2O3 films. *Mater. Today Phys.* **17**, 100356 (2021).
- [50] Chikoidze, E. *et al.* Electrical properties of p-type Zn:Ga2O3 thin films. *J. Vac. Sci. Technol., A* **40**, 043401 (2022).
- [51] Modak, S. *et al.* Variable temperature probing of minority carrier transport and optical properties in p-Ga2O3. *APL Mater.* **10**, 031106 (2022).
- [52] Lee, J. *et al.* Effect of 1.5 MeV electron irradiation on  $\beta$ -Ga2O3 carrier lifetime and diffusion length. *Appl. Phys. Lett.* **112**, 082104 (2018).

## REVIEWERS' COMMENTS

Reviewer #2 (Remarks to the Author):

Presented work is an important brick in the understanding and demonstration of great potentiality of fascinating Gallium oxide material for power electronics.

My congratulations to authors on their achievements!!

## **Response Letter (Manuscript ID: NCOMMS-23-13639A)**

We would like to thank Reviewers for their recognition of our work. All comments from reviewers have been addressed point-by-point, and there is no changes required according to the comment. In addition, we have edited the manuscript to comply with the Nature publishing policies and formatting requirements, and the revisions have been highlighted in the revised manuscript. The replies are listed below.

### *REVIEWERS' COMMENTS*

*Reviewer #2 (Remarks to the Author)*

*Presented work is an important brick in the understanding and demonstration of great potentiality of fascinating Gallium oxide material for power electronics. My congratulations to authors on their achievements!!*

**Reply:** Thanks for the reviewer for his/her recognition of this work. There is no changes required for the manuscript.